# FGF2 alters macrophage polarization, tumour immunity and growth and can be targeted during radiotherapy

Jae Hong Im[1,5], Jon N. Buzzelli[1,5], Keaton Jones[1], Fanny Franchini [2], Alex Gordon-Weeks [3], Bostjan Markelc[1], Jianzhou Chen [1], Jin Kim[4], Yunhong Cao[1] & Ruth J. Muschel [1 ✉]

Regulation of the programming of tumour-associated macrophages (TAMs) controls tumour growth and anti-tumour immunity. We examined the role of FGF2 in that regulation. Tumours in mice genetically deficient in low-molecular weight FGF2 (FGF2[LMW]) regress dependent on T cells. Yet, TAMS not T cells express FGF receptors. Bone marrow derived-macrophages from *Fgf2[LMW−/−]* mice co-injected with cancer cells reduce tumour growth and express more inflammatory cytokines. FGF2 is induced in the tumour microenvironment following fractionated radiation in murine tumours consistent with clinical reports. Combination treatment of in vivo tumours with fractionated radiation and a blocking antibody to FGF2 prolongs tumour growth delay, increases long-term survival and leads to a higher iNOS[+]/CD206[+] TAM ratio compared to irradiation alone. These studies show for the first time that FGF2 affects macrophage programming and is a critical regulator of immunity in the tumour microenvironment.

[1] Oxford Institute for Radiation Oncology, University of Oxford, Oxford OX3 7DQ, UK. [2] The Kennedy Institute of Rheumatology, Roosevelt Dr, Oxford OX3 7FY, UK. [3] Nuffield Department of Surgical Sciences, University of Oxford, Oxford OX3 9DU, UK. [4] Galaxy Biotech, 1230 Bordeaux Dr, Sunnyvale, CA 94089, USA. [5] These authors contributed equally: Jae Hong Im, Jon N. Buzzelli. ✉email: ruth.muschel@oncology.ox.ac.uk

Tumour-associated macrophages (TAMs) are highly abundant in a range of solid tumours[1]. Whilst some clinical data indicates an association with reduced patient survival[2,3], others suggest specific subsets of macrophages can improve patient outcomes[4,5]. These disparities may be due to macrophages having both pro- and anti-tumourigenic effects. Macrophage nomenclature has evolved over recent years to take account of these differing phenotypes, and although some debate still remains, it is largely accepted that most TAMs lie on a spectrum that can encompass features of both classically activated 'M1' macrophages, exhibiting immune-stimulatory functions, and alternatively activated 'M2' immune-suppressive macrophages[6–9]. Experimental evidence demonstrates that TAMs are often skewed towards a pro-tumourigenic phenotype. Tumour microenvironmental stimuli, including tumour-derived cytokines, can polarise macrophages towards an M2 phenotype[10,11], promoting tumour growth through mechanisms including angiogenesis, proliferation, local invasion, metastatic potential and immune escape[1,12]. Macrophages also mediate resistance to therapy including radiotherapy, and irradiation can augment angiogenic and pro-tumourigenic macrophage phenotypes[6,13–15]. Recent data suggest a key role for macrophages in modulating the host adaptive immune response, primarily via suppressive effects on CD8$^+$ T lymphocytes[16].

In this study, we demonstrate that fibroblast growth factor 2 (FGF2) plays a pivotal role in shifting TAMs towards a pro-tumourigenic, M2-like phenotype in the tumour microenvironment. FGF2 is one member of a family of related proteins (FGFs) that exert mitogenic activity by binding to FGF receptors (FGFR)[17–20]. Of these receptors, FGF2 has the highest affinity for FGFR1, 2 and 3b[21]. FGF2 has many isoforms, generated by the use of alternative translational start sites[22]. The lowest molecular weight isoform, FGF2$^{LMW}$, is the only secreted form[23,24], and is a highly potent angiogenic molecule[19,22,25]. FGF2$^{LMW}$ also affects the response to injury in skin, nerve and cartilage[19,20,22,26,27]. The higher molecular weight forms of FGF2 are often nuclear in location and can regulate transcription independently of FGFRs. In previous work, we found that FGF2$^{LMW}$ was highly expressed by myeloid cells in colorectal cancer (CRC) liver metastases. Blocking FGF2 with an antibody delayed growth of murine CRC liver metastases through vascular renormalisation[25].

Here, we show that tumours regressed in mice genetically deficient in FGF$^{LMW}$ (Fgf2$^{LMW−/−}$) while shifting TAM polarisation. Cancer cells failed to develop substantial liver metastases or subcutaneous tumours in Fgf2$^{LMW−/−}$ mice. Fgf2$^{LMW−/−}$ TAMs were more inflammatory, (M1-like) than in C57Bl6 wildtype (WT) mice, and subcutaneous tumour regression was T cell-dependent. In WT mice, TAMs were the major source of FGF2, and were the only immune cell to abundantly express FGFR1 and 2. FGF2 has been reported to be induced in irradiated human tumours[28–31] raising the possibility that FGF2 might be a useful therapeutic target for patients who have received radiotherapy. Accordingly, we examined the effect of FGF2 on the irradiation response, and found that a blocking antibody to FGF2 in combination with radiotherapy reduced or even eliminated tumour regrowth in association with an increase in the TAM iNOS$^+$/CD206$^+$ ratio (also called a M1/M2 ratio).

## Results

### Genetic elimination of *FGF2$^{LMW}$* in the host results in tumour regression.
To ask whether liver metastases would be constrained in Fgf2$^{LMW−/−}$ mice as they were by blocking anti-FGF2 antibody[25,32], we examined the growth of liver metastases in Fgf2$^{LMW−/−}$ mice. After intrasplenic injection of the murine CRC cell line, MC38 or the pancreatic cancer cell line, KPC into WT or Fgf2$^{LMW−/−}$ mice, Fgf2$^{LMW−/−}$ mice had significantly less liver tumour burden (Fig. 1a) with macroscopic colonies evident in only 3 of 8 (37%) and 2 of 6 (33%) Fgf2$^{LMW−/−}$ mice injected with MC38 and KPC cells respectively compared with 100% in WT mice. Histological analysis showed increased immune infiltration in Fgf2$^{LMW−/−}$ mice, particularly at the metastasis-liver border (arrows, Fig. 1b).

After subcutaneous injection MC38($1 × 10^4$ cells) and KPC ($2 × 10^5$ cells) tumours reached endpoint by day 21 in WT mice, yet only 1 of 10 (10%) MC38 and 5 of 8 (63%) KPC tumours in Fgf2$^{LMW−/−}$ mice reached endpoint by day 120 (500 mm$^3$, Fig. 1c–f). Initially tumours from both cell lines grew in Fgf2$^{LMW−/−}$ mice (albeit significantly slower than in WT mice, Fig. 1d, f), then growth plateaued, and tumours either regressed completely or resumed growth after a delay (between day 40 and 70; Fig. 1d, f). This growth pattern also occurred with a cell line of non-gastrointestinal origin, the lung carcinoma cell line, LLC in Fgf2$^{LMW−/−}$ mice (Supplementary Fig. 1a, b). Histological analysis of subcutaneous tumours revealed increased immune infiltration in tumours in Fgf2$^{LMW−/−}$ mice (Fig. 1g), similar to the liver metastasis model suggesting that FGF2$^{LMW}$ might mediate tumour immunity.

### Loss of FGF2 leads to T cell recruitment in tumours.
To investigate the immune infiltration in tumours in Fgf2$^{LMW−/−}$ mice, we collected immune cells from tumours in WT and Fgf2$^{LMW−/−}$ mice at day 10 after inoculation for analysis by flow cytometry (Fig. 1h). Tumours in Fgf2$^{LMW−/−}$ mice contained increased proportions of CD4$^+$ and CD8$^+$ T cells. The percentages of F4/80$^+$ macrophages were unaltered and there was an increase in CD11b$^+$Gr-1$^{HIGH}$ granulocytes (Fig. 1h). Immunohistochemistry confirmed that subcutaneous tumours and liver metastases in Fgf2$^{LMW−/−}$ mice had increased infiltration by T cells (CD3$^+$ cells, both CD4 and CD8), which extended into the clusters of tumour cells (Fig. 1i).

We characterised changes in cytokine and chemokine expression by qPCR of tumour lysates. Tumours from the Fgf2$^{LMW−/−}$ mice had increased levels of many CCL and CXCL chemokines, and pro-inflammatory cytokines, many characteristically expressed by myeloid cells (Supplementary Fig. 2). These data provide evidence of a shift towards the inflammatory spectrum in tumours in Fgf2$^{LMW−/−}$ mice.

### T cells are required for tumour regression in *Fgf2$^{LMW−/−}$* mice.
Because of the tumour regression and increased proportion of T cells in tumours of Fgf2$^{LMW−/−}$ mice, we asked whether depletion of T cells would rescue tumour growth in Fgf2$^{LMW−/−}$ mice (Fig. 1j). T cells were depleted using anti-CD3 antibody. In PBS treated Fgf2$^{LMW−/−}$ mice, tumours began to regress by day 8, consistent with the experiments shown above. In contrast, T cell depletion in Fgf2$^{LMW−/−}$ mice led to rapid tumour growth. Overall these studies demonstrate that T cells mediate tumour regression in Fgf2$^{LMW−/−}$ mice. However, T cell depletion resulted in less rapid tumour growth in Fgf2$^{LMW−/−}$ mice than in WT mice implicating a component of T cell independent alteration in growth (Fig. 1j).

In addition, the absence of FGF2$^{LMW}$ led to altered cytokine expression in CD4$^+$ and CD8$^+$ T cells. CD4$^+$ and CD8$^+$ T cells isolated from tumours of Fgf2$^{LMW−/−}$ mice had increased RNA expression of *Il6*, *Il12*, and *Il17* compared with T cells isolated from WT mice. CD4$^+$ T cells also had increased *Ifnγ* and CD8$^+$ T cells had increased *Il13*. Both showed enhanced expression of the proliferation marker *Ki67* (Fig. 1k). Thus, both CD4$^+$ and CD8 T cells showed evidence of increased activation and proliferation in tumours of Fgf2$^{LMW−/−}$ mice.

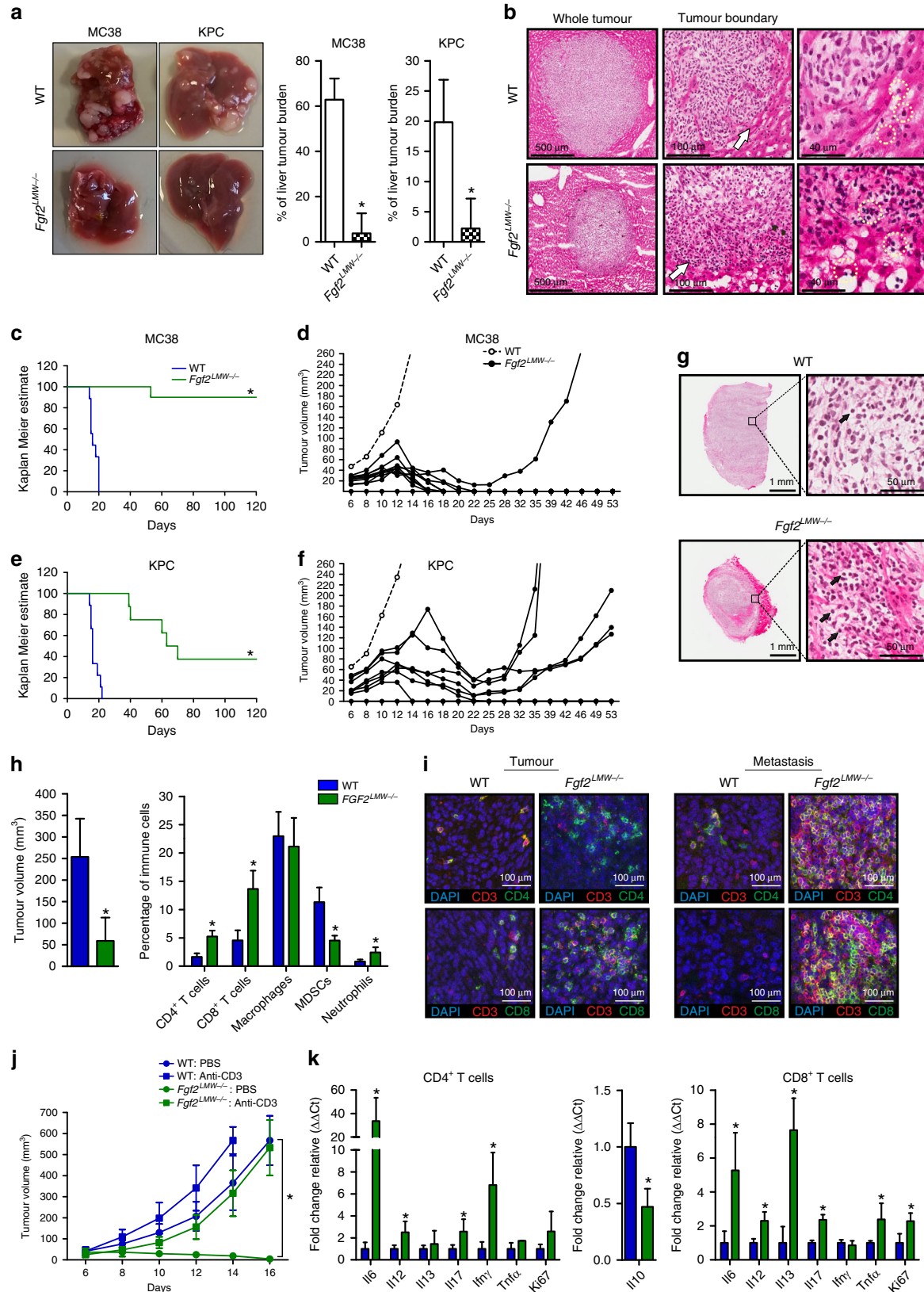

**Macrophages express FGF receptors following interaction with tumour cells**. Since an immune response was required for tumour regression in *Fgf2^LMW−/−* mice, we asked which immune cells expressed FGF2 receptors. Despite the involvement of T cells in tumour regression, we failed to detect FGFR1 or 2 on T cells or other immune cell types from either naïve mice or from tumour bearing mice, except TAMs. Over 85% of TAMs expressed FGFR1 and ~65% expressed FGFR2. Less than 5% of myeloid cells from naïve spleens expressed either receptor (Fig. 2a, b). TAMs also expressed more *Fgf1, 3 and 4 RNA* and substantially more *Fgfr2* RNA than naïve bone marrow derived-macrophages (BMDM) with over a 100-fold increase in *Fgfr*2 (Fig. 2c). Thus

**Fig. 1 Depletion of FGF2$^{LMW}$ leads to T cell mediated tumour regression. a** Macroscopic analysis of liver metastasis tumour burden following intrasplenic injection of MC38 and KPC tumour cells in WT ($n = 12$) and $Fgf2^{LMW-/-}$ ($n = 14$) mice. **b** Representative H&E images of liver metastases MC38 tumour nodules in WT and $Fgf2^{LMW-/-}$ mice at different magnifications; Arrows indicates tumour-liver interface with increased lymphoid infiltration in $Fgf2^{LMW-/-}$ mice and circles highlight lymphoid cells **c** Kaplan–Meier Estimate and (**d**) tumour growth curves following subcutaneous injection of MC38 in WT ($n = 9$) and $Fgf2^{LMW-/-}$ ($n = 9$) mice. **e** Kaplan–Meier Estimate and (**f**) tumour growth curves following subcutaneous injection of KPC in WT ($n = 9$) and $Fgf2^{LMW-/-}$ ($n = 9$) mice. **g** Representative H&E images of MC38 subcutaneous tumours in WT and $Fgf2^{LMW-/-}$ mice. Arrows indicate lymphoid cells. **h** Tumour volume and flow cytometry analysis of immune cells isolated from MC38 subcutaneous tumours in WT ($n = 12$) and $Fgf2^{LMW-/-}$ ($n = 13$) mice 10 days post-tumour cell injection. **i** Left; confocal imaging of CD3$^+$CD4$^+$ T cells and CD3$^+$CD8$^+$ T cells in MC38 subcutaneous tumours of WT and $Fgf2^{LMW-/-}$ mice 10 days post-tumour cell injection. Right; confocal imaging of CD3$^+$CD4$^+$ T cells and CD3$^+$CD8$^+$ T cells in liver metastases MC38 tumour nodules of WT and $Fgf2^{LMW-/-}$ mice 20 days post-tumour cell injection. **j** Depletion of T cells using anti-CD3 antibody (clone 17A2) following MC38 tumour inoculation in WT and $Fgf2^{LMW-/-}$ mice. Antibody was injected at day −1, day 3 and day 7. $N = 8$ for each group. **k** RNA expression in CD4$^+$ T cells and CD8$^+$ T cells isolated from subcutaneous tumours of WT ($n = 6$) and $Fgf2^{LMW-/-}$ ($n = 9$) mice using qRT-PCR analysis. * represents statistical significance compared with control mice ($p \leq 0.05$) using Kruskal–Wallis one-way ANOVA. Error bars indicate S.D.

macrophages were the main immune cell subtype within tumours that might be expected to respond directly to FGF2.

Since BMDM themselves expressed little *Fgfr* RNA, we asked whether tumour cells could directly affect FGFR expression by macrophages. We co-cultured BMDM with tumour cells for 24 h and with irradiated tumour cells, as tumour irradiation is known to influence macrophage activation[15,33–39]. Co-culture of BMDM with MC38 or KPC tumour cells led to increased expression of FGFR1 and 2 on BMDM and this increase was greater if the tumour cells had been irradiated (Fig. 2d, e). Further conditioned medium (CM) from MC38 cancer cells was sufficient to induce *Fgfr1* and *Fgfr2* although with a longer time course (Fig. 2f). We then asked whether induction of macrophage polarisation using LPS, a canonical stimulator of M1 and IL4, a stimulator of M2 polarisation induced FGFR1 and 2. LPS induced expression of FGFR1 and 2 in BMDM, however IL4 only induced FGFR1 (Fig. 2g). Thus tumour cells can influence macrophages to induce FGFR expression.

**TAMs are the primary source of FGF2 in tumours**. To identify the sources of FGF2, we performed flow cytometry analysis for FGF2 on cells isolated from the tumour. Macrophages expressed FGF2 abundantly (83.3%), while fewer MDSCs or granulocytes expressed FGF2, 9.7% and 29.9% respectively. In contrast, FGF2 was expressed in <4% of T cells (Fig. 3a). TAMS express several hundred folds more *Fgf2* RNA than BMDM. Of the cultured tumour cells, only MC38 expressed detectable *Fgf2* RNA and this was several hundred folds less than the amounts from TAMs (Fig. 3b). It should be noted that both antibody staining and RNA analysis detect FGF2 high molecular weight forms as well as FGF2$^{LMW}$. To further assess *Fgf2* expression in cancers, we analysed the relationship of *Fgf2* expression to immune sub-population markers in human CRC using GSEA analysis (Fig. 3c). Macrophage and granulocyte rich tumours, as identified by gene expression signatures, correlated with increased *Fgf2* expression, whereas CD8$^+$ T cells, CD4$^+$ T cell, Th1 cells and Th2 cells showed no correlation (Fig. 3c). These data are consistent with murine studies identifying myeloid cells as a major source of FGF2$^{LMW}$[40].

**TAMs are polarised towards an inflammatory (M1) phenotype in tumours of $Fgf2^{LMW-/-}$ mice**. A shift in TAM polarisation, without altered TAM number, can have profound effects on tumour growth[16,41]. Therefore, we assessed TAM polarisation in $Fgf2^{LMW-/-}$ mice by flow cytometry and RNA analysis (Fig. 4a). iNOS is well described as a marker of M1 polarisation[42], while CD206 is an M2 marker[42]. TAMs from $Fgf2^{LMW-/-}$ mice had a significant increase in iNOS expression compared with TAMs from WT mice, while over 90% expressed CD206 in both

genotypes illustrating the complexity of macrophage polarization (Fig. 4a). We then isolated TAMs from WT and $Fgf2^{LMW-/-}$ mice 10 days after tumour cell inoculation and analysed RNA expression (Fig. 4b). TAMs isolated from $Fgf2^{LMW-/-}$ mice had a significant increase in M1 or inflammatory markers compared with those from WT mice, including *Il6, Ifnγ, Tnfα and iNos*, and *H2-Eα* (*MHC II*) (Fig. 4b). TAMs isolated from $Fgf2^{LMW-/-}$ mice also had decreased expression of the M2 markers, *Ym1* and *Ym2* (Fig. 4b). These data demonstrate marked alterations in the polarisation of TAMs from tumours in $Fgf2^{LMW-/-}$ mice compared with those from WT mice.

We evaluated cytokine induction after exposure of BMDM from $Fgf2^{LMW-/-}$ or WT mice to polarisation stimuli, LPS (M1) or IL4 (M2). Without stimulation, $Fgf2^{LMW-/-}$ BMDM had higher levels of the pro-inflammatory cytokines, *Cxcl1, Il1β, Il6* and *Tnfα*, and decreased *Ym2*, a M2 marker compared with WT BMDM (Fig. 4c). Following LPS stimulation, increases in *Cxcl1, Cxcl2, Il1β* and *Il6* RNA expression in $Fgf2^{LMW-/-}$ BMDM and decreases in *Ym1* and *Ym2* were similar in both genotypes (Fig. 4c). IL4 treatment led to a reduction of *iNos* in $Fgf2^{LMW-/-}$ BMDM and WT BMDM, and an increase in *Ym1* and *Ym2* (Fig. 4c). Thus the overall levels reflected the presence or absence of FGF2$^{LMW}$, but the changes after stimulation were similar for both.

We asked whether exposure to FGF2 in culture altered macrophage cytokine expression. Addition of rFGF2 to BMDM from either WT or $Fgf2^{LMW-/-}$ mice resulted in induction of *Cxcl1,2* and *Nos2* (Supplementary Fig. 3a, b). Because exposure of BMDM to CM induced FGFRs, we then added rFGF2 after exposure of WT or $Fgf2^{LMW-/-}$ BMDM to CM, resulting in the induction of *Cxcl1,2*, and *Ym1,2* (Supplementary Fig. 3a, b). While the results in tissue culture not surprisingly do not fully replicate the expression patterns in vivo, these experiments suggest that FGF2 alters the phenotypes of TAMs, and may be critical in allowing macrophages to generate a pro-tumour response.

**$Fgf2^{LMW-/-}$ BMDM delay tumour growth**. To ask whether macrophages affect tumour growth, we co-injected BMDM with tumour cells (Supplementary Fig. 4a). BMDM co-injected with MC38 cells subcutaneously did not alter the tumour growth rate. However, co-culture of BMDM with MC38 cells prior to injection, led to a significant and sustained increase in tumour growth (Fig. 5a). Co-culture of BMDM with tumour cells led to increased expression of iNOS and CD206, and *Il1α, Il1β, Il6* and *Tgfβ* in the BMDM (Fig. 5b; Supplementary Fig. 4b). To determine whether FGF2$^{LMW}$ deficiency in BMDM influenced tumour growth, we co-injected tumour cells with uneducated WT or $Fgf2^{LMW-/-}$ BMDM subcutaneously. We used two different ratios of tumour cells to BMDM to examine dose response. Co-injection of tumour

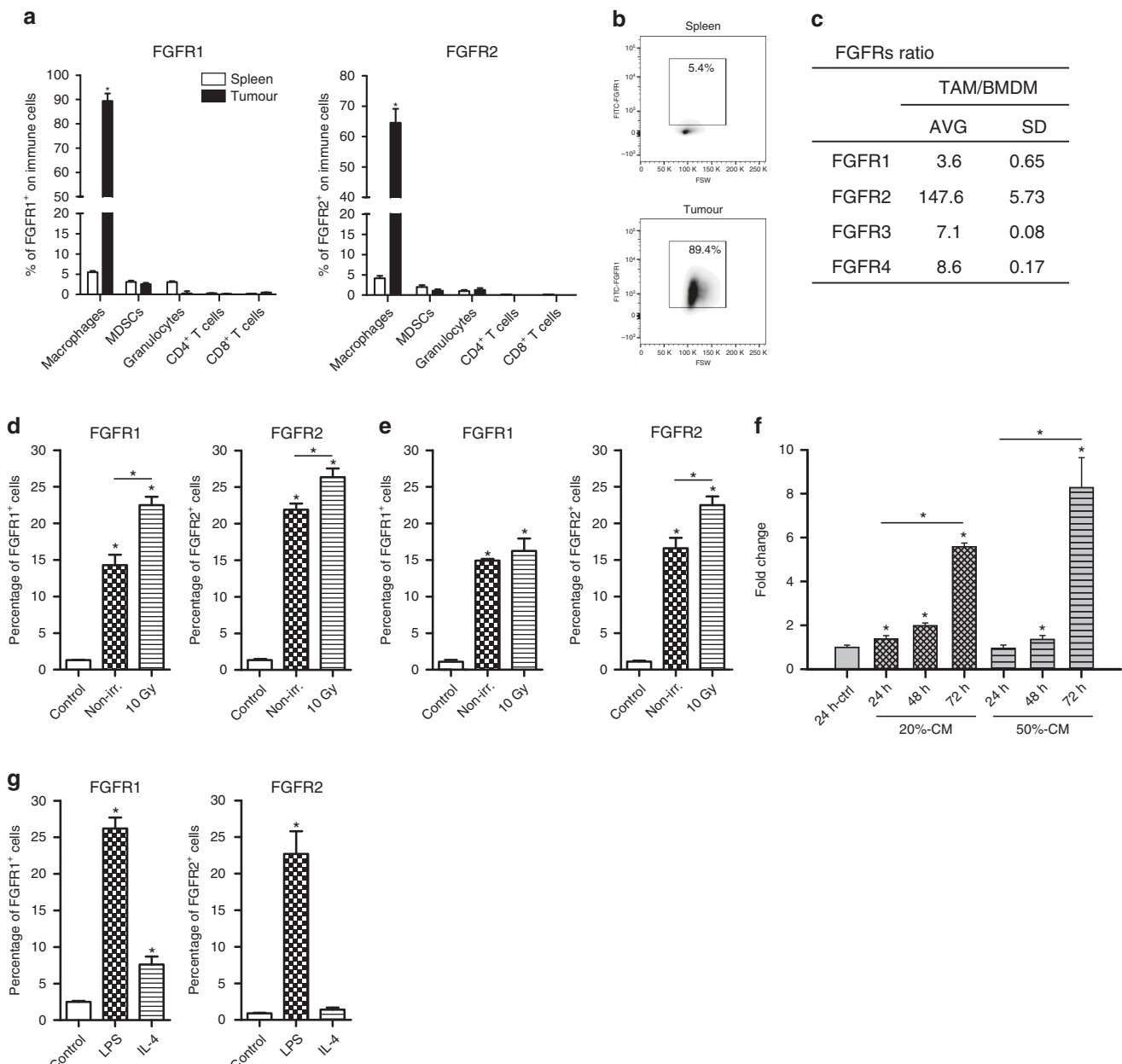

**Fig. 2 Tumour-associated macrophages express FGF receptors. a** Analysis of expression of FGFR1 and FGFR2 on immune cells from the spleen (open bars) and MC38 subcutaneous tumours (filled bars) using flow cytometry and (**b**) representative flow cytometry plots for FGFR1 in the spleen and tumour. **c** Compares RNA for FGFRs1-4 from TAMS to BMDM using qPCR. **d** FGFR1 and FGFR2 expression in WT BMDM macrophages following 24 h of co-culture with non-irradiated or irradiated MC38 tumours cells. **e** FGFR1 and FGFR2 expression in WT BMDM macrophages following 24 h of co-culture with non-irradiated or irradiated KPC tumours cells. **f** Fold change in FGFR1 RNA in BMDM after 24 h exposure to MC38 conditioned medium. **g** FGFR1 and FGFR2 expression in WT BMDM macrophages following stimulation with LPS or IL4. $N = 6$ in each group.

cells with WT BMDM did not affect tumour growth at a 1:1 or 1:4 ratio (Fig. 5c). In contrast, co-injection of $Fgf2^{LMW-/-}$ BMDM with tumour cells delayed tumour growth in a dose-dependent manner (Fig. 5c, d). Thus macrophage genotype and conditioning affected tumour growth.

To ask whether these macrophages altered the immune infiltrate, we collected the subcutaneous tumours 14 days post-inoculation and analysed the tumour infiltrate (Fig. 5e). Compared with control tumours, co-injection with $Fgf2^{LMW-/-}$ BMDM resulted in an increase in the proportion of CD8[+] T cells, whereas co-injection with WT BMDM did not alter the tumour infiltrate (Fig. 5e). To assess the contribution of an adaptive immune response, we repeated our co-injection

experiments (1:1 ratio) in immunodeficient SCID mice, which lack mature T and B cells (Fig. 5f). Following co-injection of $Fgf2^{LMW-/-}$ BMDM with tumour cells in immunodeficient SCID mice, we observed a similar tumour growth delay, demonstrating that the tumour growth delay resulting from $Fgf2^{LMW-/-}$ BMDMs is in part independent of T cells (Fig. 5f).

**FGF2 and the response to radiation therapy.** FGF2 is only sporadically reported as increased in human cancers, but radiotherapy has been shown to increase expression of $FGF2$ in human rectal and cervical cancers[30,43,44]. To determine whether this was the case in our models, we assessed FGF2[LMW] expression by Western blotting of lysates from irradiated subcutaneous tumours

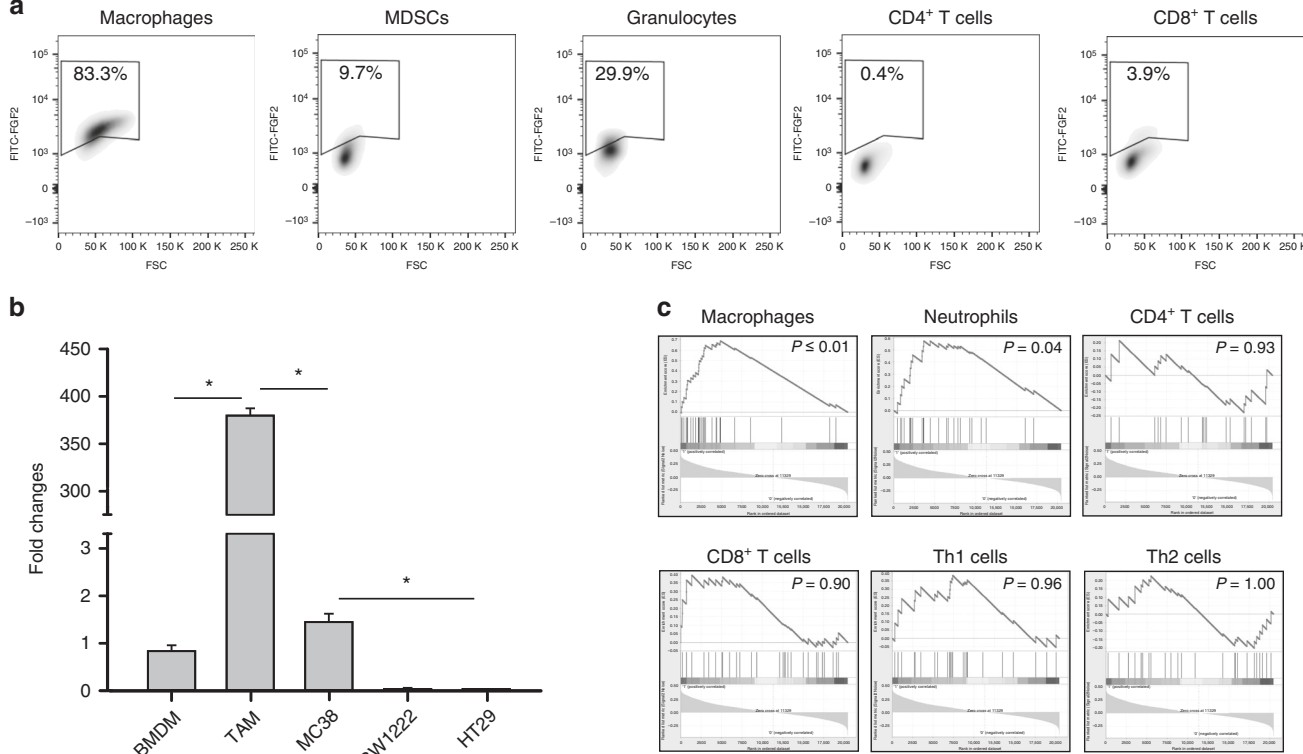

**Fig. 3 Tumour-associated macrophages express FGF2. a** Analysis of FGF2 expression in immune cells isolated from MC38 tumours of WT mice using flow cytometry; Macrophages, MDSCs, Granulocytes, CD4[+] T cells and CD8[+] T cells. **b** Shows the fold change in *Fgf2* RNA expression in TAMs isolated from MC38 tumours and from the cell lines MC38, SW1222 and HT29 using BMDM as the baseline (**c**) GSEA correlation analysis of immune cell infiltrate and FGF2 expression in colorectal cancer patients; Macrophages, Neutrophils, CD4[+] T cells, CD8[+] T cells, Th1 cells and Th2 cells. * represents statistical significance compared with control mice ($p \leq 0.05$). Error bars indicate S.D.

derived from CRC cell lines; human SW1222 and HT29, and murine MC38. We used fractionated radiation at 2 Gy in analogy to the treatment of human cancers, although at lower total doses. Following fractionated radiation of nine fractions of 2 Gy, the amount of FGF2[LMW] in all three tumour models was increased (Fig. 6a, b). There are no antibodies that distinguish FGF2[LMW] from FGF2[HMW] because all FGF2[HMW] include the entire low-molecular weight protein. Thus immunostaining will capture FGF2[HMW]. Recognising this important caveat, that staining could also indicate FGF2[HMW], we applied immunohistochemistry to locate and quantitate the FGF2 in the irradiated tumours (Fig. 6c, d). MC38 FGF2 staining intensity was increased in irradiated tumours compared with unirradiated (Fig. 6c). In both irradiated and unirradiated tumours FGF2 staining was noted in most F4/80[+] macrophages, further suggesting that TAMs are a major source of FGF2. A similar pattern was seen in SW1222 tumours (Fig. 6c). In both cases staining was seen in other cell types as well as in areas that are likely extracellular. HT29 had a higher level of FGF2 staining in cell types other than F4/80 positive cells and there was less increase in overall FGF2 staining after irradiation, although absolute levels were high and most of the F4/80 positive staining areas also stained for FGF (Fig. 6c, d). The staining in other cells might reflect FGF2[HMW] expression in tumour cells. MC38 tumours in *Fgf2*[LMW−/−] mice, control or irradiated failed to show any FGF2 staining of tumour cells or of macrophages further suggesting that the contribution of the tumour cells to FGF2 is minimal for MC38 cells (Fig. 6e). Thus, as reported in human rectal and cervical cancers[28–31], radiotherapy led to increased total tumour content of FGF2[LMW], and FGF2 was found in TAMs.

**Diminished FGF2 or FGF2 blocking antibody improves survival following radiotherapy.** In Fig. 1 tumours generated after a subcutaneous injection of $1 \times 10^4$ MC38 cells resulted in an immune response leading to decreased growth and rejection. Because higher numbers of inoculated cells can result in diminished anti-tumour immune responses[45] we injected $2 \times 10^5$ MC38 cells and did not see rejection or even differences in tumour growth in WT and *Fgf2*[LMW−/−] mice (Fig. 7a, Supplementary Fig. 5a). We then used these tumours to examine the response to radiation therapy. Tumours in *Fgf2*[LMW−/−] mice were substantially more responsive to radiation than the tumours in WT mice with greater growth delays and a higher percentage of cure (Fig. 7a, Supplementary Fig. 5a).

We asked whether an anti-FGF2 blocking antibody would similarly affect the tumour response to radiotherapy. MC38 tumours in WT mice received fractionated radiation of $9 \times 2$ Gy after reaching 100 mm$^3$. Anti-FGF2 blocking antibody was administered every second day for 4 weeks. The anti-FGF2 antibody itself did not affect tumour growth in unirradiated mice. Mice whose tumours received radiation reached endpoint 12 days post-irradiation, while the addition of anti-FGF2 antibody led to a significant growth delay with mice reaching the experimental endpoint on average 24 days post-irradiation (Fig. 7b; Supplementary Fig. 5b). Thus similar to the results with tumours in *Fgf2*[LMW−/−] mice, the addition of anti-FGF2 blocking antibody delayed growth of irradiated MC38 tumours.

We asked what effect these treatments might have on human tumour cell lines. We used both a radiation sensitive tumour from SW1222 cells and a more resistant cell line HT29 in immuno-suppressed mice administering the same radiation dose as above.

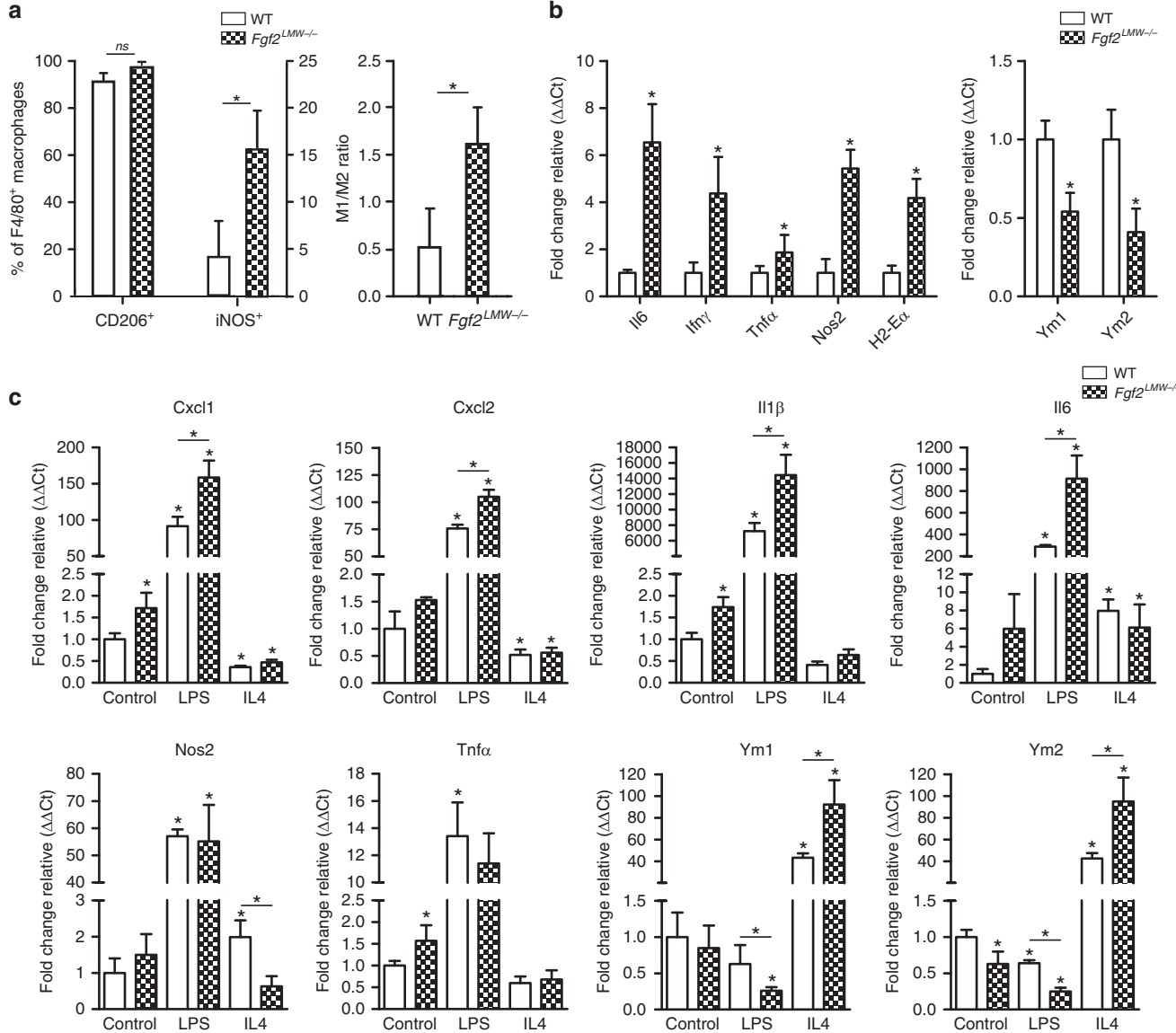

**Fig. 4 *Fgf2^LMW−/−* macrophages are polarised towards an M1 phenotype. a** Macrophage analysis of iNOS and CD206 in WT (*n* = 6) and *Fgf2^LMW−/−* (*n* = 9) mice 10 days post-tumour cell injection using flow cytometry. **b** RNA expression analysis of macrophages isolated from subcutaneous tumours of WT and *Fgf2^LMW−/−* mice 10 days post-tumour cell injection. **c** RNA expression in WT and *Fgf2^LMW−/−* BMDM following stimulation with LPS or IL4 using qRT-PCR analysis; *Cxcl1, Cxcl2, Il1β, Il6, iNOS, Tnfα, Ym1* and *Ym2*. * represents statistical significance compared with control mice (*p* ≤ 0.05). Error bars indicate S.D.

In both cases administration of the anti-FGF2 antibody had no effect on tumour growth in unirradiated mice. However, addition of the anti-FGF2 blocking antibody significantly enhanced the efficacy of irradiation (Fig. 7c, d; Supplementary Fig. 5c, d). For SW1222 tumours, the more radiosensitive tumours, 9 of 11 mice that received combination therapy survived for over 170 days, and 6 of 11 mice had complete tumour regression by 210 days, compared with 1 of 8 mice in the irradiated group (Fig. 7c; Supplementary Fig. 5c). For HT29, irradiation alone significantly delayed tumour growth, with mice reaching endpoint 35 days post-treatment, compared with 40 days for mice receiving combination therapy, a significant but smaller prolongation (Fig. 7d Supplementary Fig. 5d). Administration of the anti-FGF2 blocking antibody also greatly delayed tumour growth of irradiated MC38 tumours in syngeneic, immunocompetent mice. Thus, in both immunosuppressed and immunocompetent models given the same dose to more or less radiosensitive models, addition of anti-FGF2 blocking antibody during radiotherapy

enhanced the growth delay and increased the incidence of long-term survival.

We asked whether we could demonstrate FGF2 mediated autocrine growth factor effects. Addition of antibody to FGF2 or a pan-FGFR inhibitor (BGJ398) did not reduce cellular viability, proliferation or clonogenic survival after radiation (Supplementary Figs. 6, 7a, b). Further even though the cancer cell lines used in this study variably expressed FGFR1-4 (Supplementary Fig. 7c), neither BGJ398 nor recombinant FGF2 in tissue culture altered the best described downstream mediators of FGFR signalling, pERK and pAKT (Supplementary Fig. 7d, e). Thus we failed to find evidence of FGF2 mediated autocrine growth factor signalling by the tumour cell lines that we used.

We asked whether the FGF2 blocking antibody led to effects on tumour vascularity. Here the data were model specific. FGF2 blocking antibody reduced vascularity slightly in the MC38 model, but not in the SW1222 model (Supplementary Fig. 8). In neither case did it affect tumour growth (Fig. 7a–c) or

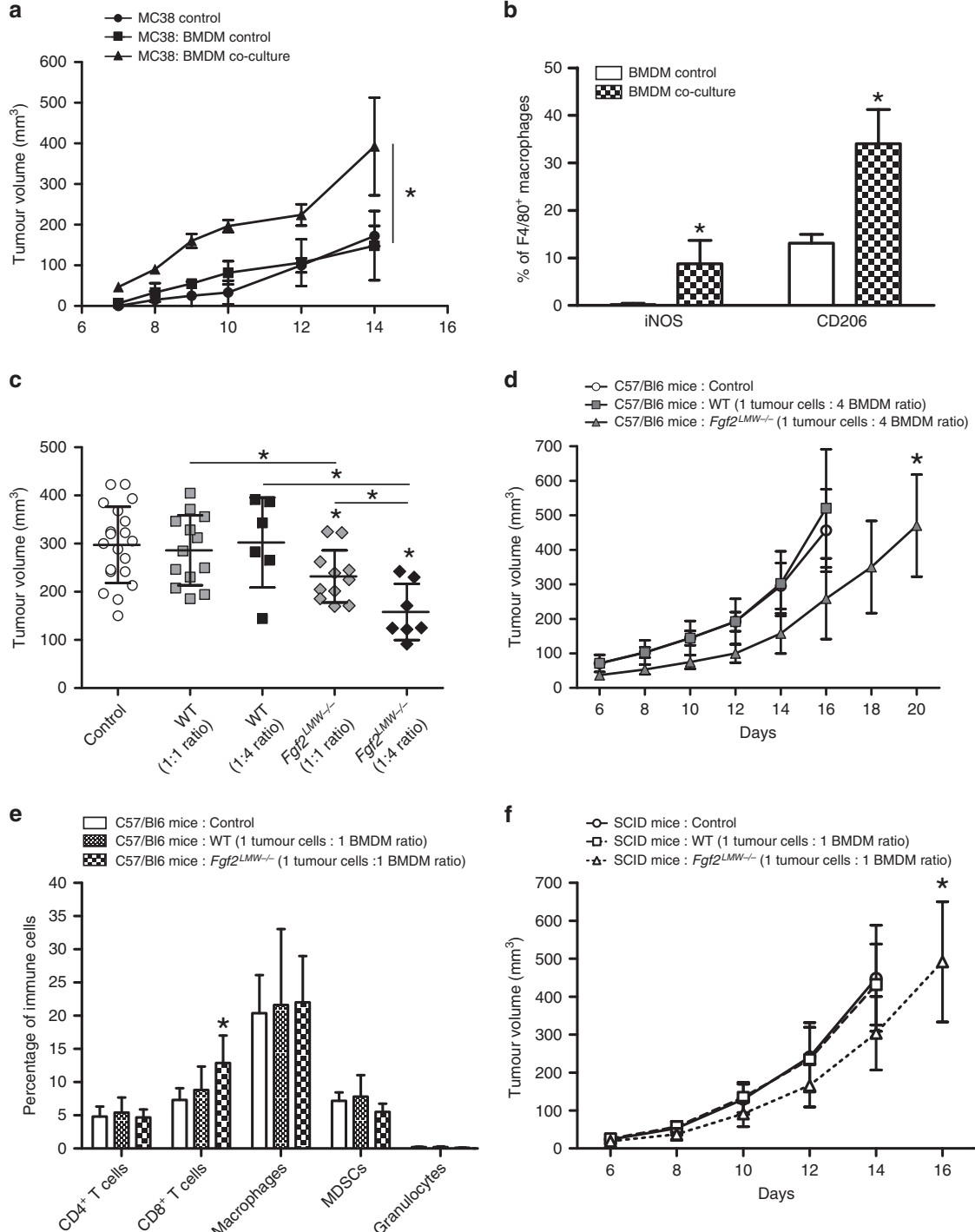

**Fig. 5 Co-injection of *Fgf2^{LMW−/−}* BMDM with tumour cells delays tumour growth. a** Growth curves following subcutaneous co-injection of MC38 tumours cells and WT BMDM pre-educated with MC38 tumour cells. **b** iNOS and CD206 expression in BMDM following pre-education with MC38 tumour cells. **c** Tumour volume 14 days post co-injection of MC38 tumour cell with WT or *Fgf2^{LMW−/−}* BMDM at different concentrations. **d** Growth curves following subcutaneous co-injection of MC38 with either WT or *Fgf2^{LMW−/−}* BMDM at a ratio of 1 tumour cells: 4 BMDM in C57Bl6 WT mice. **e** Flow cytometry analysis of immune cells isolated from WT BMDM and *Fgf2^{LMW−/−}* BMDM subcutaneous tumours of C57Bl6 mice 14 days post-tumour cell injection. **f** Growth curve analysis following subcutaneous co-injection of MC38 with either WT BMDM or *Fgf2^{LMW−/−}* BMDM at a ratio of 1 tumour cells: 1 BMDM in SCID mice. * represents statistical significance compared with control mice ($p \leq 0.05$). Error bars indicate S.D. $N = 10$ in each group.

distribution of vessel size (Supplementary Fig. 8b). Radiation reduced vascularity in both models. Vascular size was notably reduced by FGF2 blocking antibody in irradiated SW1222 but not in MC38 tumours (Supplementary Fig. 8c). When using a

different measure, perfused vessel volume, to evaluate vascularity, MC38 tumour vascularity was unaffected by anti-FGF2 antibody, but was reduced after irradiation. Further tumours in *Fgf2^{LMW−/−}* mice had approximately the same vascular density

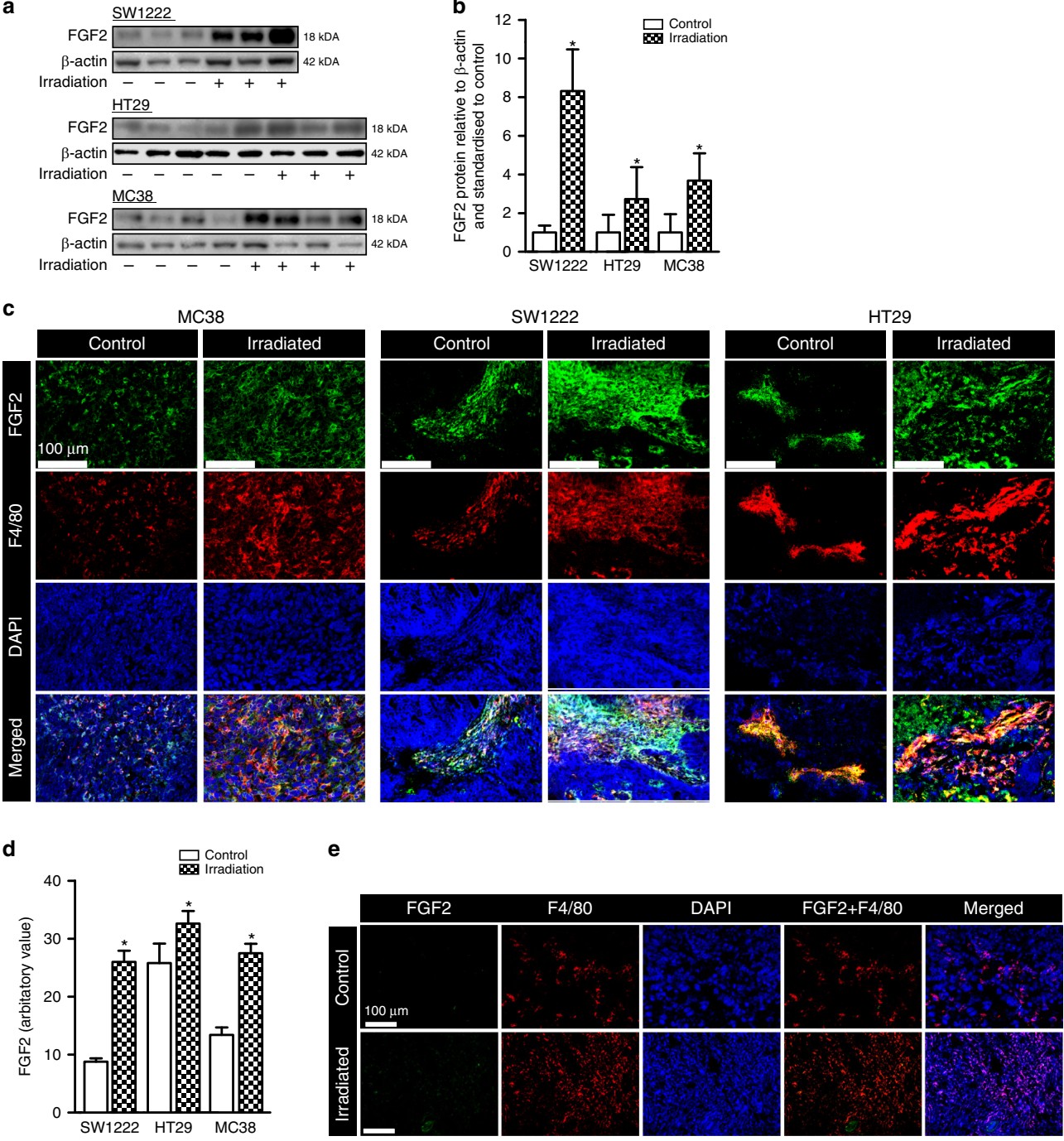

**Fig. 6 Effect of Irradiation on FGF2 in tumours. a** Representative western blot images of FGF2 in SW1222, HT29 and MC38 tumours following irradiation. **b** Quantification of Western blot images. **c** Representative images of F4/80 and FGF2 confocal imaging in MC38, SW1222 and HT29 tumours following irradiation. **d** Quantification of FGF2 from immunofluorescent images in (**c**) following irradiation. **e** Representative images of FGF2, F4/80, DAPI and combined images of FGF2 and F4/80 or all signals merged from immunohistochemistry and confocal imaging of MC38 tumours grown in syngeneic mice. * represents statistical significance compared with control mice ($p \leq 0.05$). Error bars indicate S.D.

as tumours in WT mice. Irradiation reduced the vascular density in $Fgf2^{LMW-/-}$ mice and vascular volume (Supplementary Fig. 9a, b). Thus whilst anti-angiogeneic effects might play a role, this effect did not appear to be substantial or generalisable.

**Irradiation and anti-FGF2 antibody combination therapy promotes TAM polarisation.** Consistent with many reports, we found irradiated tumours had higher proportions of CD11b+F4/80+ TAMs than controls (Fig. 8a, c). Administration of anti-

FGF2 antibody marginally reduced (without statistical significance) the proportion of macrophages in the non-irradiated tumours, but did not alter the increase in CD11b+F4/80+ cells following irradiation (Fig. 8a–c). Although anti-FGF2 antibody did not influence the proportion of TAMs, it had a profound effect on macrophage polarisation after irradiation (Fig. 8d–f). The iNOS+/CD206+ (M1/M2) ratio of TAMs decreased with irradiation in the SW1222 tumours, but was unchanged in the other two models (Fig. 8d–f). The effect of the anti-FGF2

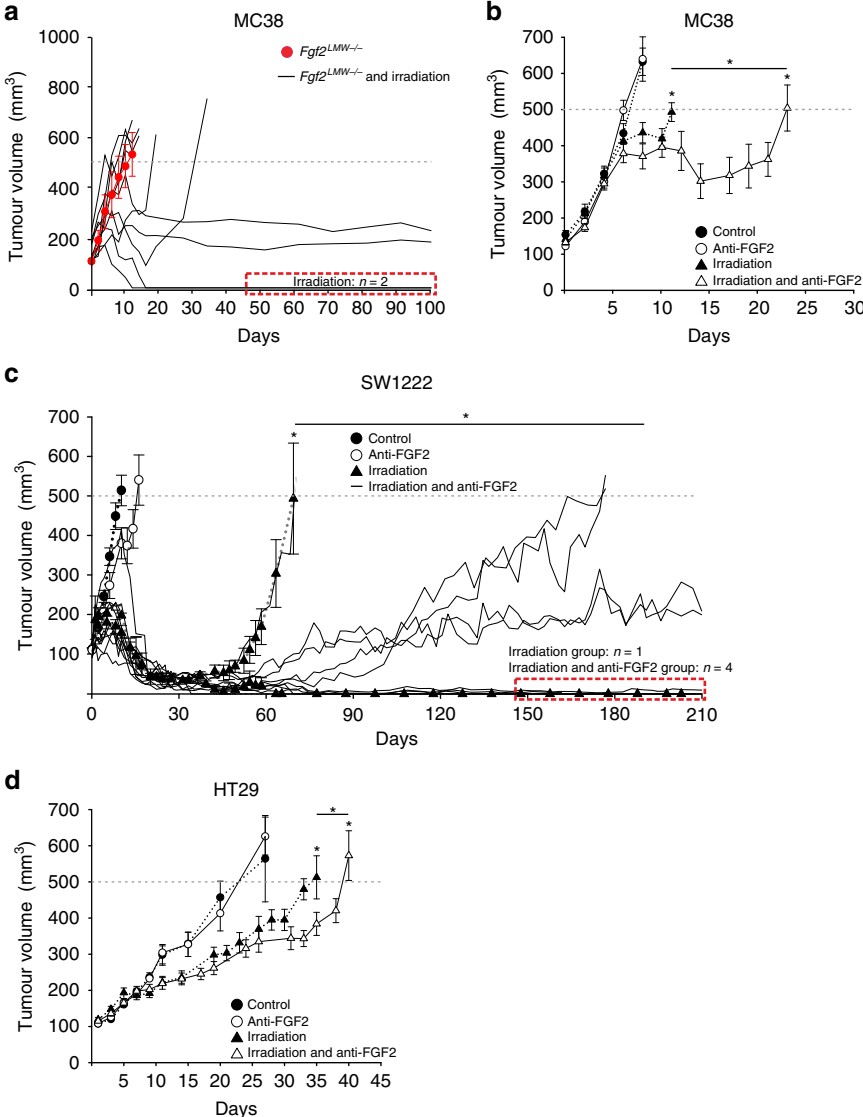

**Fig. 7 Effect of irradiation and reduced FGF2 on tumour growth.** Growth curves of (**a**) MC38 tumours in $Fgf2^{LMW-/-}$ mice (**b**) MC38 tumours in C57Bl6 mice (**c**) SW1222 in nude mice (**d**) HT29 in nude mice following anti-FGF2 antibody, irradiation or combination therapy as indicated. N is shown at Supplementary Fig. 5E. * represents statistical significance compared with control mice ($p \leq 0.05$). Error bars indicate S.D.

antibody alone was also variable. However, combination of the radiotherapy and anti-FGF2 antibody consistently led to an increased iNOS+/CD206+ macrophage ratio (Fig. 8d–f). Whilst MC38 cells were tested in syngeneic mice, the combination of irradiation and anti-FGF2 antibody also effectively prolonged survival of immunosuppressed mice bearing SW1222 and HT29 subcutaneous tumours. Thus, blocking FGF2 led to changes in TAM polarisation towards the M1 spectrum and correlated with enhanced growth delay after radiotherapy.

## Discussion
Tumour-associated macrophages (TAMs) have the capacity for both pro- and anti-tumourigenic functions[46,47]. Here we show that FGF2 in the tumour microenvironment can be a potent factor in directing macrophages towards a more pro-tumourigenic state. FGF2 in cancer has previously been implicated as an angiogenic factor[20,25,48] with the caveat that increased FGF2 is not always associated with increased vascularity[49]. FGF2 has not previously been implicated in the regulation of macrophages, nor in tumour immunity[25,48].

In our experiments, cancer cells including MC38 at lower innocula failed to generate continued tumour growth in $Fgf2^{LMW-/-}$ mice with tumour regression dependent on T cells. In these tumours TAMs were the predominant immune cells that expressed FGF2 and its receptors. Although the proportion of macrophages infiltrating the tumours was equivalent in WT and $Fgf2^{LMW-/-}$ mice, the nature of these TAMs was substantially different. Macrophage polarisation or activation can lead to a complex variation of expression patterns and functions[9,46,47]. These range from inflammatory, anti-tumourigenic phenotypes (M1) to pro-tumourigenic and immunosuppressive functionality (M2). In general, TAMs have been described as more M2 in nature[9,16,50]. TAMs from $Fgf2^{LMW-/-}$ mice expressed many characteristic pro-inflammatory cytokines and markers compared with WT TAMs, and $Fgf2^{LMW-/-}$ BMDM suppressed tumour growth after co-injection. This suppressive effect was associated with enhanced infiltration of T cells in immunocompetent mice, but was also evident in immunosuppressed mice. This enforces the concept that FGF2 mediates tumour growth through alteration of TAMs, both dependent and independent of adaptive

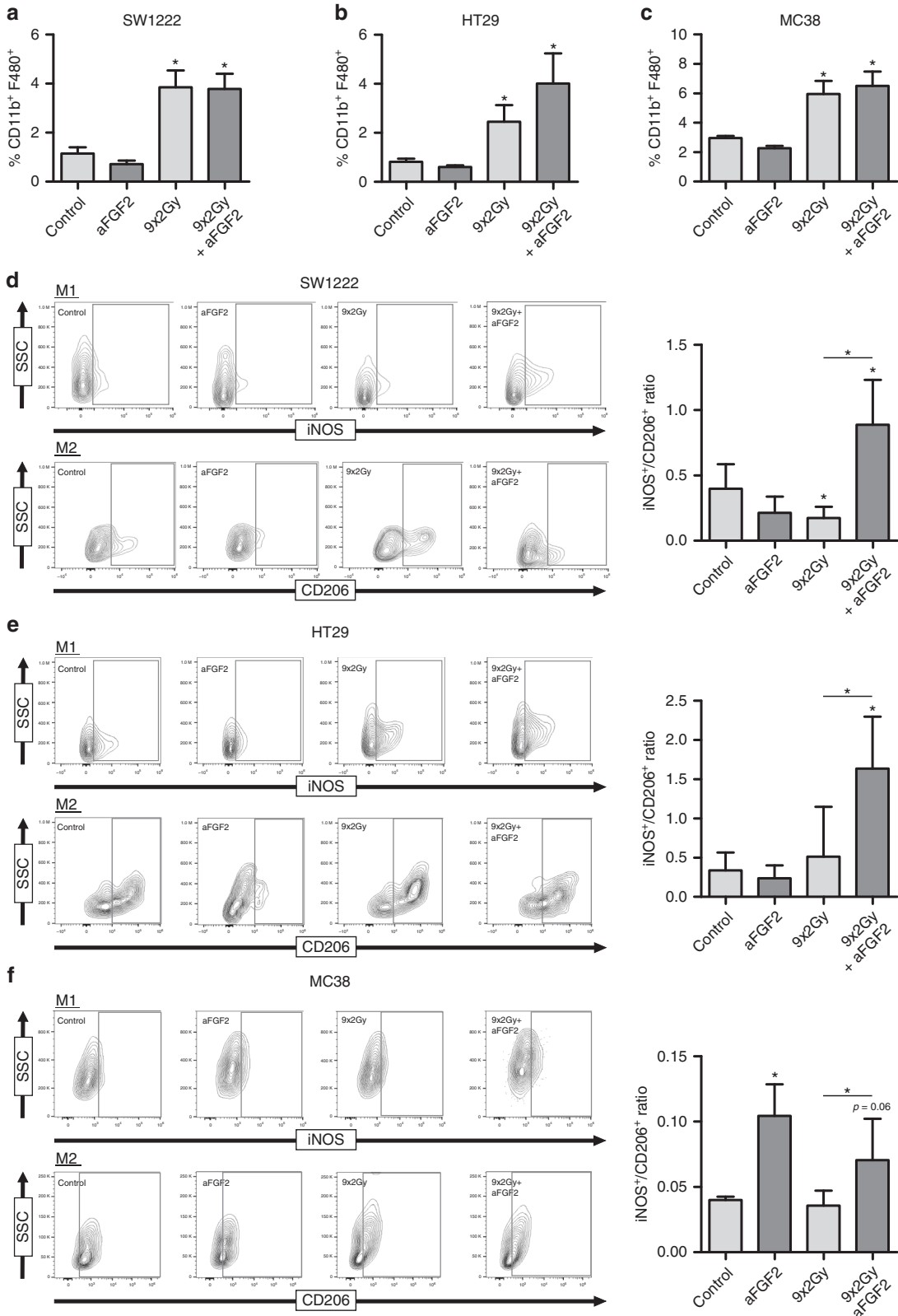

**Fig. 8 Macrophage recruitment and polarisation following irradiation and anti-FGF2 combination therapy.** Macrophage recruitment following anti-FGF2, irradiation or combination therapy for (**a**) SW1222, (**b**) HT29, and (**c**) MC38. Macrophage iNOS and CD206 expression following anti-FGF2, irradiation or combination therapy for (**d**) SW1222, (**e**) HT29, and (**f**) MC38. N is shown at Supplementary Fig. 5E. * represents statistical significance compared with control mice ($p \leq 0.05$). Error bars indicate S.D.

immunity. In addition, other cell types in the tumour microenvironment, such as fibroblasts are also likely to be affected by the $Fgf2^{LMW-/-}$ genotype and the altered phenotype of TAMs almost certainly includes indirect effects.

Macrophages have multiple effects on tumour growth, some of which can involve tumour immunity, with or without an adaptive immune response. For example, CSF1 which acts as both a chemotactic and survival factor for macrophages also promotes M2-like polarisation[51]. As a result, in some glioblastoma models, inhibition of CSF1 led to a shift in TAM polarisation away from pro-tumourigenic phenotypes resulting in tumour regression independent of adaptive immunity[51]. In other models, inhibition of CSF1 resulted in an enhanced immune response[52]. Furthermore, inhibition of PI3Kγ blocked M2 polarisation of macrophages, promoted T cell recruitment, and enhanced anti-tumour immunity[16,41]. Here we showed that co-injection of BMDM conditioned by tumour cells had a tumour growth promoting effect. We also found that BMDM deficient in $Fgf2^{LMW-/-}$ mice have an anti-tumour effect. These actions were both adaptive immune dependent and independent.

Radiotherapy is known to stimulate adaptive immunity to tumours[53–55], but nonetheless an effective anti-tumour response often fails to occur even after administration of immune checkpoint therapy[56]. Macrophage recruitment and polarisation after irradiation of tumours is one factor contributing to immunosuppression, which can counteract the immune stimulatory effect of irradiation[15,57]. It should be noted that alteration of macrophage recruitment and polarization has been shown to radiosensitize tumours. Agents such as anti-CSFR1 antibody and SDF1 inhibition have been used in this way[35,38,39,58]. The dose of radiation makes a difference[37,59]. Klug et al. showed that at relatively low doses, up to 2 Gy, radiation led to improved T cell infiltration into tumours due to induction of NOS2 expressing macrophages[37]. Some of these effects were due directly to irradiation of the macrophages. In our hands at higher doses a similar macrophage phenotype also was influenced by the absence or blocking of FGF2. We observed an increase in FGF2 following irradiation of tumours in mice similar to those described in several clinical reports[28–30,43,44]. Blocking FGF2 with a specific antibody led to diminished tumour regrowth after irradiation that was associated with an increase in M1 polarisation of TAMs. It should be noted that there are at least 22 human FGFs. Blocking only FGF2 is likely to have very different effects than blocking less specifically the FGF-FGFR signalling pathways. Treatment with the blocking anti-FGF2 antibody did not affect tumour growth or macrophage polarisation unless irradiation had been applied, similar to the tumours formed from high innocula in $Fgf2^{LMW-/-}$ mice. We failed to find evidence that FGF2 was acting as an autocrine growth factor for these cancer cells. Macrophages influence tumour angiogenesis[40]. FGF2 certainly might affect angiogenesis directly in these models, but the effect could also be indirect through reprogramming of macrophages. In our hands blocking FGF2 or its absence did not have a generalisable effect on tumour vascular density before or after radiation. Inhibition of tumour growth after irradiation occurred in both immunocompetent and immunosuppressed mice. The extent of the additional effect of the anti-FGF2 antibody correlated with the extent of the tumour response to radiation alone. These data are consistent with previous reports showing inhibition or depletion of macrophages after irradiation leads to prolonged tumour growth delay or in some cases augments immune checkpoint therapy[35,38,60,61].

Here, we directly associate TAMs with increased tumour growth based on experiments showing enhanced tumour growth after co-injection of cancer cells with pre-conditioned BMDMs. Macrophages were the only immune cell population in tumours

to abundantly expression FGFR1 or 2, and in tissue culture, we showed that tumour cells can induce the expression of FGFR1 and 2 in BMDM. Furthermore, FGF2 levels are increased in the tumour microenvironment following irradiation in vivo. Collectively, these findings suggest that TAMs in the tumour microenvironment are ready to engage with FGF2 in the irradiated tumour microenvironment leading to reprogramming to generate a more pro-tumourigenic milieu. These TAMs can then contribute to tumour growth in both an immune dependent and independent fashion.

Our work suggests that macrophage polarisation could be a component driving delayed tumour growth and tumour regression in $Fgf2^{LMW-/-}$ mice. It also suggests that FGF2 in the tumour microenvironment is an important regulator of macrophage differentiation, particularly during radiotherapy. Further recognition of factors, which contribute to TAM polarisation could lead to novel therapeutic strategies. In non-cancer models, effects on macrophages by FGF2 have not been directly described. However, enhanced leucocyte recruitment directed by FGF2[62], and promotion of autoimmune arthritis[61] are both suggestive of potential macrophage involvement. FGF2 deficiency leads to enhanced colitis in murine models[63], demonstrating the capacity of FGF2 to mediate inflammation. Furthermore, FGF2 supports wound healing, a process associated with a rapid influx of myeloid cells[64], and depletion of FGF2 delays wound healing[65]. Even though effects on macrophages have not been described directly in these settings, our work introduces these possibilities and raises additional questions about effects on fibroblasts, other cell types and other more complex indirect actions. Overall our work defines FGF2 as a key modulator of macrophage polarisation in tumours and demonstrates that FGF2, as well as other factors which regulate macrophage activity, can be targeted during radiotherapy.

## Methods
All methods are available in the Supplementary Information methods section.

**Reporting summary**. Further information on research design is available in the Nature Research Reporting Summary linked to this article.

## Data availability
The data that support the findings of this study are available from the corresponding author upon reasonable request. Images of the gels used in figures are available in the Supplementary Information.

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

## Acknowledgements

Funding by Cancer Research UK, C5255/A23755. The authors would like to thank Cary Queen, Galaxy Biotech, Sunnyvale, CA for his interest and help in this work.

## Author contributions

J.I.: conceptualisation, methodology, investigation and revision writing; J.N.B.: conceptualisation, methodology, investigation, and original draft writing; K.J.: methodology, investigation, and resources; F.F.: methodology, and resources; A.G.W.: methodology; B.M.: methodology; J.C.: methodology; J.K.: methodology; YC: methodology; RJM: conceptualisation, resources, original and revision writing, project administration, and funding acquisition.

## Competing interests

The authors declare the following competing interests: Jin Kim is the Chief Scientific Officer of Galaxy Biotec, which supplied the FGF2 blocking antibody, GAL-F2 used in this study.
