## [Peer Review File · Nature Communications]

Reviewers' Comments:

Reviewer #1:

Remarks to the Author:

The manuscript by Buzzelli et al. focuses on understanding the contributions of host-derived FGF2LMW to tumor growth and macrophage polarization using various tumor models. Initial studies demonstrate a significant decrease in tumor growth following subcutaneous injection into FGF2LMW^{-/-} mice using various models (MC38, KPC and LLC). Further studies demonstrate alterations in immune profiles favoring an increased anti-tumor response in the FGF2^{-/-}LMW mice driven by T cells. In addition, macrophages isolated from control and FGF2^{-/-}LMW mice show altered expression of various cytokines and chemokines. To further define the contributions of macrophages to this phenotype, tumor cells were co-injected with control or FGF2^{-/-}LMW bone marrow derived macrophages, which demonstrates reduced tumor growth although in a T cell-independent manner. Finally, irradiation was found to promote increased levels of FGF2, and treatment with anti-FGF2 led to better response to irradiation, which correlated with an increased iNOS:CD206 ratio in macrophages. Based on these findings, the authors conclude that FGF2 regulates tumor immunity and growth through altering macrophage polarization. While the link between FGFs and immune modulation within the tumor microenvironment is not necessarily novel, the findings that macrophage-derived FGF2 impacts tumor growth and that anti-FGF2 enhances efficacy of irradiation are of interest. While these studies demonstrate a role for host-derived FGF2LMW in promoting tumor growth, and effectively demonstrate efficacy of anti-FGF2 in the context of irradiation, some concerns are noted as described below.

The authors should provide information in the Materials and Methods regarding the specific substrains of C57BL/6 mice that were used for these studies. As written, it appears that the FGF2LMW^{-/-} mice were purchased from Jackson Laboratories on the C57BL/6J background and maintained in-house, and the control mice were purchased from Charles River, which provides the C57BL/6N substrain. If this is correct, this should be noted in the methods section. Furthermore, in light of recent observations that differences in substrain can impact immune phenotypes (for example, PMID 27210752), if different substrains were indeed used in these studies, this should be discussed as a potential caveat particularly of the immune phenotype analysis. Ideally, key experiments involving immune profiling, including the macrophage polarization studies, should be repeated using mice of the same substrain.

Figure 3: While basal levels of a number of the factors shown are significantly different than control, the magnitude of induction seems to be similar. Does restoration of FGF2 to the FGF2LMW^{-/-} cells impact basal expression levels of these factors?

Figure 5C: The conclusion that the majority of FGF2-producing cells appear to be TAMs is not supported by the images provided as there are substantial areas of green FGF2 staining that do not appear to be associated with F4/80 staining. Quantification of dual-staining or flow cytometry-based quantification should be provided to support these conclusions.

The authors conclude several times (in the title, lines 215-216 and 352) that "macrophage polarization by FGF2LMW mediates tumor growth". However, this conclusion seems premature given the studies provided. There are several other possible explanations for the phenotypes that have not been ruled out, such as reduced tumor growth due to loss of macrophage-derived FGF2-mediated effects on tumor cells, or other non-immune host cells, which could ultimately impact both tumor growth rates and the immune environment. Thus, additional potential explanations for the phenotypes observed should be thoroughly considered.

The authors demonstrate that expression levels of FGFR1 and FGFR2 are higher on TAMs compared with other immune cell populations. Does FGF2 directly activate FGFR signaling in TAMs?

In Figure 1, the authors demonstrate that the reduction in tumor growth observed in the

FGF2LMW^{-/-} mice is T cell-dependent. However, in Figure 4, T cells are found to be indispensable for the reduction in tumor growth induced by the FGF2LMW^{-/-} macrophages. Potential reasons for these differences (such as the possibility that non-macrophage FGF2LMW-expressing host cells are impacting the T cell population in the FGF2LMW^{-/-} mice) should be discussed.

Because the flow plots are presented in the context of FSC or SSC, the authors should provide the gating schemes used to identify the different immune cell populations (as Supplementary data).

Please provide the number of mice used for each in vivo experiment.

Figure 1: Figures J and K are switched when referred to in the text.

Figure 2: Please clarify whether this antibody recognizes only the secreted form of FGF2, or whether it also recognizes the intracellular form of FGF2, and whether this impacts interpretation of the data.

Figure 3A: The shifts are difficult to see in the panels provided and may be more easily interpreted as histograms. Also, please provide the iNOS:CD206 ratios along with statistical analysis.

Figure 4E: The authors conclude that there is a reduction in MDSCs in tumors with FGF2LMW^{-/-} macrophages, although this does not appear to be supported by the data presented in the graph.

Figure 6: For clarity, please provide the cell number count for each plot.

Although a general statement regarding statistical analysis is included in the materials and methods section, it is not clear which test was used for each graph, which could be provided in the Figure Legends.

Reviewer #3:

Remarks to the Author:

Buzzelli et al present experimental evidence that suggests a role of FGF2 as a key modulator of macrophage polarization in tumors, as part of the "wound response" associated with localized radio-therapy. They identify FGF2 as a potential target of translational interest, since antibodies against FGF2 could remove one of radio-therapy immune-suppressive effects, while exploiting other pro-immunogenic ones.

Several concerns:

1. The mechanism of radio-therapy induction of FGF2 is not clear. In vitro experiments are needed to specifically define the nature of signaling after irradiation, whether it is different in different cells types and whether it is irradiation-does dependent.
2. Is FGF2 induction part of DDR? A heat map for FGF2 null versus parental lines could help understanding the pathway. It could also provide important hints regarding the component of cross talk with innate immunity
3. The previous work of the authors in reference 25 shows an effect of FGF2 on angiogenesis, independent of irradiation. Understanding the mechanism of FGF2 induction by irradiation is key, since it may also have clinical impact in cancer metastasis.
4. A main concern is the use of only one radiotherapy regimen (9GyX2), without clear justification for this selection. Particularly in the in vivo experiments a different dose could have overcome the progression of tumors and/or the type of TAMs infiltration. Since no dose response curves were provided in the model used, one can not exclude that re-population because of inadequate dose may have also recruited M2 and/or induced polarization of TAMs. The choice of this radio-therapy

dose and number of fractions needs a much more explicit justification, since others have shown very different results for instance when low doses are used. (Klug et al Cancer Cell. 2013 Nov 11;24(5):589-602)

5. Figure 5 Shows modest difference in HT29 model and the delay in MC38 is only transient. From the images it is hard to conclude a sustained superior effect by the combination

6. References on irradiation and macrophages are incomplete (missing key papers of Deng L et al, Xu J et al etc.).

Reviewer #4:

Remarks to the Author:

In this manuscript, Buzzelli et al. show slower growth of primary tumors and decreased liver metastasis in FGF2^{l/w} deleted mice. Mechanistic studies showed that the deletion induces T cell recruitment into the tumors and that this is critical for the decreased tumor growth. Although the authors were not able to detect FGFR1 or 2 in the T cells, they found that the majority of tumor-associated macrophages expressed FGFR1 and 2. Both FGF2^{l/w} deletion and anti-FGF2 antibody induced TAM polarization towards a more inflammatory phenotype. Decreased tumor xenograft growth was seen in nude mice when anti-FGF2 blocking antibody was used in combination with irradiation. The manuscript reports many interesting, but somewhat dissociated results that raise several questions.

In the previous paper by the same authors, metastasis-associated neutrophils expressed substantially more fibroblast growth factor 2 (FGF2) in comparison to naïve neutrophils, indicating neutrophil polarization by the tumor microenvironment. Administration of FGF2 neutralizing antibody to mice bearing experimental liver metastases phenocopied neutrophil depletion by reducing liver metastatic colony growth, vascular density, and branching (Hepatology 2017;65:1920-1935).

It has also been shown that FGFR1 inhibition by gene silencing in tumor cells or administration of a compound (SSR128129E at nanomolar concentrations) inhibiting FGF-induced signaling in tumor xenografts increases the efficiency of radiotherapy (pubmed/26896280).

Thus, an alternative possibility to explain the authors' results is that FGF2^{l/w} depletion directly causes initial TC death in the tumors *in vivo*, and that the resulting danger signals invite inflammatory/immune cells to the tumors. This possibility is consistent with the "T-cell independent alteration" that "resulted in less rapid tumor growth in FGF2^{l/w} mice than in WT mice". Although the authors show that anti-FGF2 antibody treatment after irradiation did not alter clonogenic survival of SW1222 and HT29 cells, it would be essential to also analyze the effect of FGF2 silencing.

In the manuscript, a limited analysis is reported on the role of TC produced FGF, and the expression of FGF receptors. A comparison of the amounts of FGF2 produced by TCs and macrophages should be shown. A more careful mapping of the autocrine pathway in the TCs is also needed. It would be essential to compare the survival of the control, FGF2 and FGFR1 silenced as well as FGF2 antibody treated tumor cell cultures after irradiation.

On pages 5 and 6 the authors show that T-cells are required for tumor regression in the FGF2^{l/w} mice. However, later, in Figure 5 F-H and Supplementary Figure 4, they show that T-cells are not required for the effects of the anti-FGF2 antibody on human SW1222 and HT29 tumor xenografts in immunodeficient mice. The authors should analyze the mouse tumor cell lines in using irradiation with and without the anti-FGF2 antibody in an immunocompromised background. Alternatively, they could use another inhibitor of the FGF2-FGFR pathway, such as SSR128129E, a FGFR1 blocking antibody, soluble receptor or FGFR1 gene silencing in the tumor cells. In addition, the authors should discuss why anti-FGF2 as a monotherapy does not decrease xenograft growth

(Fig 5 F-H), although FGF2^{l/w} deletion decreases tumor growth in the isogenic mice (Fig. 1).

As the authors indicate in the introduction, FGF2^{l/w} is a highly potent angiogenic molecule. However, here the authors don't provide any tumor vessel analysis. This should be analyzed to exclude the possible role of decreased vascular density to the decreased tumor growth. If anti-FGF2 antibody treatment decreases vascular density as shown previously in the case of liver metastasis, it could explain why anti-FGF2 antibody treatment decreases SW1222 and HT29 tumor growth in immunodeficient mice.

Specific comments:

Another question concerns the specificity of the effects reported here. The authors should perhaps mention that the other 21 members of FGF family are not expressed or of concern here. Similarly, they should describe which of the four receptors are present in the tumor cells they use.

When reporting immune infiltration to the metastases/tumors, the authors should also report if this concerned normal tissues or circulating immune cells.

The authors should re-isolate the escape clones that resumed growth and test if they regrow in corresponding experiments in FGF2^{l/w} mice.

The LLC growth curves should be added to Supplementary Figure 1.

Did TAMs express FGFR3 or FGFR4? Was phosphorylation of FGFR1 correlated with FGF2^{l/w} expression or irradiation in TCs or TAMs?

The anti-FGF2 antibody was used for the MC38 clonogenic assay in Supplementary Figure 5. However, only human tumor cells were used in the in vivo experiments. Why were the mouse tumor cells not used in the in vivo tumor experiments?

As the authors explain, FGF2 expression has been reported to be induced in irradiated human tumors. Thus, Figs 5A and B are of little value. In Fig. 5E, much of the macrophage and FGF2 staining appears to be eradicated by the irradiation. The authors should show the specificity of anti-FGF2 by staining macrophages of the corresponding tumors grown in FGF2^{l/w} mice.

Minor comments

- Fig. 1 J and K are mixed on page 6?
- How was liver tumor burden measured?
- How was tumor volume calculated? The authors should study M1/M2 ratio in tumors beyond the (very small) volume of 500mm³.
- Fig. 1 B: Please show the invasion front in high magnification, with the identification of the tumor cells (by IHC).
- In all experiments, the number of mice should be mentioned in each group. The individual tumor growth curves are difficult to interpret, for example in Fig. 5F.

Response to reviewers' comments:

Reviewer #1: Cancer immunology (TAMS)
(Remarks to the Author):

The manuscript by Buzzelli et al. focuses on understanding the contributions of host-derived FGF2^{LMW} to tumor growth and macrophage polarization using various tumor models. Initial studies demonstrate a significant decrease in tumor growth following subcutaneous injection into FGF2^{LMW}^{-/-} mice using various models (MC38, KPC and LLC). Further studies demonstrate alterations in immune profiles favoring an increased anti-tumor response in the FGF2^{-/-}LMW mice driven by T cells. In addition, macrophages isolated from control and FGF2^{-/-}LMW mice show altered expression of various cytokines and chemokines. To further define the contributions of macrophages to this phenotype, tumor cells were co-injected with control or FGF2^{-/-}LMW bone marrow derived macrophages, which demonstrates reduced tumor growth although in a T cell-independent manner. Finally, irradiation was found to promote increased levels of FGF2, and treatment with anti-FGF2 led to better response to irradiation, which correlated with an increased iNOS:CD206 ratio in macrophages. Based on these findings, the authors conclude that FGF2 regulates tumor immunity and growth through altering macrophage polarization. While the link between FGFs and immune modulation within the tumor microenvironment is not necessarily novel, the findings that macrophage-derived FGF2 impacts tumor growth and that anti-FGF2 enhances efficacy of irradiation are of interest. While these studies demonstrate a role for host-derived FGF2^{LMW} in promoting tumor growth, and effectively demonstrate efficacy of anti-FGF2 in the context of irradiation, some concerns are noted as described below.

The authors should provide information in the Materials and Methods regarding the specific substrains of C57BL/6 mice that were used for these studies. As written, it appears that the FGF2^{LMW}^{-/-} mice were purchased from Jackson Laboratories on the C57BL/6J background and maintained in-house, and the control mice were purchased from Charles River, which provides the C57BL/6N substrain. If this is correct, this should be noted in the methods section. Furthermore, in light of recent observations that differences in substrain can impact immune phenotypes (for example, PMID 27210752), if different substrains were indeed used in these studies, this should be discussed as a potential caveat particularly of the immune phenotype analysis. Ideally, key experiments involving immune profiling, including the macrophage polarization studies, should be repeated using mice of the same substrain.

We agree and were aware of this important issue, but did not include the appropriate details in the previous submission. We thank the reviewer for this comment so that we can now clarify that the FGF2^{LMW}^{-/-} mice were backcrossed with the C57BL/6N mice. We have changed the text in the Supplementary Methods to read “C57BL/6N, SCID and Athymic nude mice were purchased from Charles River Laboratories. FGF2^{LMW}^{-/-} mice were purchased from the Jackson Laboratory (FGF2^{tm2Doe}/J; Stock no: 010698; Jackson Laboratory, USA), and were bred in-house on a C57BL/6N background for 5 generations prior to use.”

Figure 3: While basal levels of a number of the factors shown are significantly different than control, the magnitude of induction seems to be similar.

We agree that the basal levels are different but the levels of induction are similar and the text has been changed to make this clear. “Thus the overall levels reflected the presence or absence of FGF2^{LMW}, but the changes after stimulation were similar for both.”

Does restoration of FGF2 to the FGF2^{LMW}^{-/-} cells impact basal expression levels of these factors?

We added recombinant FGF2 (rFGF2) to cultures of BMDM from both WT and *Fgf2*^{LMW}^{-/-} mice and found increased levels of expression in both of some macrophage markers, *Cxcl1,2* and *Ym1,2*.

This data is shown in Supplementary Fig. 3. Thus rFGF2 does alter the basal expression levels of some of the cytokines expressed by BMDM in tissue culture. We note in the text that “While the results in tissue culture not surprisingly do not fully replicate the expression patterns in vivo, these experiments suggest that FGF2 alters the phenotypes of TAMs, and may be critical in allowing macrophages to generate a pro-tumour response.”

The conclusion that the majority of FGF2-producing cells appear to be TAMs is not supported by the images provided as there are substantial areas of green FGF2 staining that do not appear to be associated with F4/80 staining. Quantification of dual-staining or flow cytometry-based quantification should be provided to support these conclusions.

We have addressed this issue in two ways. First we stained MC38 tumours grown in *Fgf2^{LMW-/-}* mice with antibody to FGF2. If considerable amounts of FGF2 were present in tumour cells, we would have expected staining in these tumour cells which was not apparent (Fig 6D). As expected no staining was apparent in the F4/80 positive cells. We have also selected different immunohistochemical images at higher magnification to more clearly show FGF2 and F4/80 staining (Fig 6B). The preponderance of FGF2 staining cells in MC38 and SW1222 tumours are also F4/80 positive. We agree with the reviewer that this is not the case for HT29 tumours in which many non F4/80 positive cells are also stained. We also performed co-localization analysis which is subject to many assumptions, but showed that 60-90% of the F4/80 + cells also were FGF2 positive. Since FGF2 is sequestered by the extracellular matrix, it would not be expected that all FGF2 staining would be seen in F4/80 positive cells even if F4/80 cells were the sole source of FGF2. Approximately 30-50% of all FGF2 positive staining colocalized with F4/80 positive areas. This data is shown in Supplemental Fig. 4D. The section headed “*FGF2 and the response to radiation therapy*” has been extensively rewritten to incorporate these data.

The authors conclude several times (in the title, lines 215-216 and 352) that “macrophage polarization by FGF2LMW mediates tumor growth”. However, this conclusion seems premature given the studies provided. There are several other possible explanations for the phenotypes that have not been ruled out, such as reduced tumor growth due to loss of macrophage-derived FGF2-mediated effects on tumor cells, or other non-immune host cells, which could ultimately impact both tumor growth rates and the immune environment. Thus, additional potential explanations for the phenotypes observed should be thoroughly considered.

We agree and have changed the title to read “ FGF2 Alters Macrophage Polarization, Tumour Immunity and Growth and can be Targeted during Radiotherapy.” The other sentences have been removed. We also point out in the discussion “Additionally other cell types in the tumour microenvironment, such as fibroblasts are also likely to be affected by the *Fgf2^{LMW-/-}* genotype and the altered phenotype of TAMs almost certainly includes indirect effects.”
“ and later “Even though effects on macrophages have not been described directly in these settings, our work raises these possibilities and raises additional questions about effects on fibroblasts and other more complex indirect actions.”

The authors demonstrate that expression levels of FGFR1 and FGFR2 are higher on TAMs compared with other immune cell populations. Does FGF2 directly activate FGFR signaling in TAMs?

To answer this question we exposed BMDM from both WT and *Fgf2^{LMW-/-}* mice to rFGF2 and examined the resulting expression of cytokines. As was noted above this led to increased expression of *Cxcl1, 2* and *Ym1, 2* but not consistently of *Nos2* (Supplementary Fig. 3) In that series of experiments, we also asked whether exposure of the BMDM to conditioned medium would alter

their responses since conditioned medium induced FGFRs on WT BMDM. In general the combination of conditioned medium with rFGF2 led to greater induction of these cytokines. Because the BMDM especially after exposure to conditioned medium can express FGF2, we performed an additional experiment where we added blocking antibody to FGF2 after exposure to conditioned medium and then overrode the effect of the antibody with further addition of FGF2. This led to increased expression of *Cxcl1, 2* but did not generally affect expression of *Ym1,2* or *Nos2* (Supplementary Fig. 3).

In Figure 1, the authors demonstrate that the reduction in tumor growth observed in the *Fgf2^{LMW/-}* mice is T cell-dependent. However, in Figure 4, T cells are found to be indispensable for the reduction in tumor growth induced by the *Fgf2^{LMW/-}* macrophages. Potential reasons for these differences (such as the possibility that non-macrophage FGF2LMW-expressing host cells are impacting the T cell population in the *FGF2LMW/-* mice) should be discussed.

We agree that there are a number of possible additional explanations for these effects. We have added the following to the discussion: “Additionally other cell types in the tumour microenvironment, such as fibroblasts are also likely to be affected by the *Fgf2^{LMW/-}* genotype and the altered phenotype of TAMs almost certainly includes indirect effects.” And also “Even though effects on macrophages have not been described directly in these settings, our work raises these possibilities and raises additional questions about effects on fibroblasts, other cell types and more complex indirect actions.” We also added the following “While we found some situations, such as lower initial inoculum led to FGF2 affecting T cell immunity, in other cases, after radiation, T cell mediated killing was a lesser factor. Thus FGF2 alterations in TAMs can contribute to tumour growth in both immune dependent and independent fashions.”

Because the flow plots are presented in the context of FSC or SSC, the authors should provide the gating schemes used to identify the different immune cell populations (as Supplementary data).

The same gating strategy shown with flow plots was previously published in Jones et al. (Radiation combined with macrophage depletion promotes adaptive immunity and potentiates checkpoint blockade. EMBO Mol. Med. 2018). We now cite that paper in the methods.

Please provide the number of mice used for each in vivo experiment.

This has been done.

Figure 1: Figures J and K are switched when referred to in the text.

This has been corrected. J and K are now indicated as Ji and Jii.

Figure 2: Please clarify whether this antibody recognizes only the secreted form of FGF2, or whether it also recognizes the intracellular form of FGF2, and whether this impacts interpretation of the data.

The antibody recognizes all forms of FGF2 as will any antibody that recognizes FGF2^{LMW}. We have added text to clarify this point and how it impacts on the interpretation of immunohistochemistry. “There are no antibodies that distinguish FGF2^{HMW} from FGF2^{LMW} because all FGF2^{HMW} include the entire low molecular weight protein. Thus immunostaining will capture FGF2^{HMW}. Recognizing this important caveat, we applied immunohistochemistry to locate and quantitate the FGF2 in the irradiated tumours (Fig. 6B&C).”

Figure 3A: The shifts are difficult to see in the panels provided and may be more easily interpreted as histograms. Also, please provide the iNOS:CD206 ratios along with statistical analysis.

This data has been presented now in graphical form with statistical significance indicated in what is now Fig. 4A.

Figure 4E: The authors conclude that there is a reduction in MDSCs in tumors with FGF2LMW^{-/-} macrophages, although this does not appear to be supported by the data presented in the graph.

We agree and have removed this data. We also note that MDSCs are difficult to identify by surface markers alone.

Figure 6: For clarity, please provide the cell number count for each plot.

This is now indicated in the methods. It ranged from 25,000 to 50,000 for each plot.

Although a general statement regarding statistical analysis is included in the materials and methods section, it is not clear which test was used for each graph, which could be provided in the Figure Legends.

This has been done.

Reviewer #3 : Cancer Radiotherapy

(Remarks to the Author):

Buzzelli et al present experimental evidence that suggests a role of FGF2 as a key modulator of macrophage polarization in tumors, as part of the "wound response" associated with localized radio-therapy. They identify FGF2 as a potential target of translational interest, since antibodies against FGF2 could remove one of radio-therapy immune-suppressive effects, while exploiting other pro-immunogenic ones.

Several concerns:

1. The mechanism of radio-therapy induction of FGF2 is not clear. In vitro experiments are needed to specifically define the nature of signaling after irradiation, whether it is different in different cells types and whether it is irradiation-does dependent.

We think that the current data would suggest that the increase is due to the increased influx of macrophages, which are in most of the tumours examined are the primary, but not exclusive source of FGF2. However we have not proven this hypothesis and on the advice of the editors, we are not pursuing this point here.

2. Is FGF2 induction part of DDR? A heat map for FGF2 null versus parental lines could help understanding the pathway. It could also provide important hints regarding the component of cross talk with innate immunity.

We believe this is beyond the scope of the current manuscript. Further since the tumour cells may not be the main source of the FGF2, this approach may not be helpful. See for example the case in Fig. 6D when we stained for FGF2 in MC38 tumours in *Fgf2*^{LMW^{-/-}} mice, there was little or no staining in tumour cells or host cells. Additionally since FGF2 null lines would also lack FGF2^{HMW} this would not be a simple experiment. This is because the distinction between the high molecular

weight and low molecular weight forms is due to the use of alternate translation initiation sites in the same FGF2 mRNA. Thus engineering a null cell for FGF2^{LMW} is not straightforward.

3. The previous work of the authors in reference 25 shows an effect of FGF2 on angiogenesis, independent of irradiation. Understanding the mechanism of FGF2 induction by irradiation is key, since it may also have clinical impact in cancer metastasis.

We have examined the effect of FGF2 on vascular density both with and without radiation as suggested by the reviewer. Supplementary Fig. 8 shows these data. In SW1222 there was a small, but significant decrease in vascular density after treatment of mice bearing these tumours with the blocking antibody to FGF2. Radiation led to a more substantial decrease in vascular density. Radiation plus the antibody led to no further decrease. In the MC38 model the tumours from mice that received antibody to FGF2 showed no decrease in vascular density (in contrast to what was seen in liver metastases). Radiation led to a substantial decrease, but again the combination did not lead to any significant change from radiation alone. Finally the vascular density of the MC38 tumours in the *Fgf2*^{LMW^{-/-}} mice was not reduced compared to those in WT mice. Thus effects on angiogenesis were not generalizable.

4. A main concern is the use of only one radiotherapy regimen (9GyX2), without clear justification for this selection. Particularly in the in vivo experiments a different dose could have overcome the progression of tumors and/or the type of TAMs infiltration. Since no dose response curves were provided in the model used, one can not exclude that re-population because of inadequate dose may have also recruited M2 and/or induced polarization of TAMs. The choice of this radio-therapy dose and number of fractions needs a much more explicit justification, since others have shown very different results for instance when low doses are used. (Klug et al Cancer Cell. 2013 Nov 11;24(5):589-602)

We chose 9X 2Gy for several reasons. First we wished to approximate the irradiation schedule used in cancer treatment in man which commonly involves multiple fractions usually of approximately 2 Gy. The upper limit for the number of treatments on our animal license allowed for 9 total radiation doses. For these reasons we used 9 X 2Gy recognizing that some of the tumor types, we used were much more responsive to this dose than others. We could not give higher numbers of doses in 2 Gy fractionation that might have had a greater curative effect because of this constraint on our use of animals. We would point out however that significant prolongation of growth delay and of Kaplan Meier survival estimates were obtained by the addition of the antibody to the radiation in each model regardless of the extent of response.

We completely agree that much lower doses have different effects. We now cite Klug et al. “Klug et al showed that at relatively low doses, up to 2 Gy, radiation led to improved T cell infiltration into tumours due to induction of NOS2 expressing macrophages⁵⁶. Some of these effects were due directly to irradiation of the macrophages.”

5. Figure 5 Shows modest difference in HT29 model and the delay in MC38 is only transient. From the images it is hard to conclude a sustained superior effect by the combination

We agree that in the cases in which the radiation alone had the least effect, adding FGF2 inhibition was not as effective. However as indicated above regardless the antibody still had a significant effect. We also note that that in the *Fgf2*^{LMW^{-/-}} mice the effect of radiation was greater than using antibody. Thus not surprisingly genetic elimination has a greater effect than administration of the antibody. We do not have sufficient antibody remaining to ask how its effects may vary with dose

or with the dose of radiation. It is also interesting to note that the less responsive tumour type was that with the least induction of FGF2 by radiation.

6. References on irradiation and macrophages are incomplete (missing key papers of Deng L et al, Xu J et al etc.).

We thank the reviewer for pointing out that additional references would be helpful. We have added the following: “It should be noted that alteration of macrophage recruitment and polarization has been shown to radiosensitize tumours. Agents such as anti-CSFR1 antibody and SDF1 inhibition have been used in this way ^{35,38,39,56}. The dose of radiation makes a difference ^{37,57}. Klug et al showed that at relatively low doses, up to 2 Gy, radiation led to improved T cell infiltration into tumours due to induction of NOS2 expressing macrophages ³⁷. Some of these effects were due directly to irradiation of the macrophages.” Instead of citing Deng et al directly we referenced Brown, Recht and Strober Clin Ca Res. 2017, reference 16 which carefully reviews many of the contributions in this field from the Brown lab including that of Deng et al. Xu et al was cited and is included here.

Reviewer #4 : Cancer therapy

(Remarks to the Author):

In this manuscript, Buzzelli et al. show slower growth of primary tumors and decreased liver metastasis in FGF2^{low} deleted mice. Mechanistic studies showed that the deletion induces T cell recruitment into the tumors and that this is critical for the decreased tumor growth. Although the authors were not able to detect FGFR1 or 2 in the T cells, they found that the majority of tumor-associated macrophages expressed FGFR1 and 2. Both FGF2^{low} deletion and anti-FGF2 antibody induced TAM polarization towards a more inflammatory phenotype. Decreased tumor xenograft growth was seen in nude mice when anti-FGF2 blocking antibody was used in combination with irradiation. The manuscript reports many interesting, but somewhat dissociated results that raise several questions.

In the previous paper by the same authors, metastasis-associated neutrophils expressed substantially more fibroblast growth factor 2 (FGF2) in comparison to naïve neutrophils, indicating neutrophil polarization by the tumor microenvironment. Administration of FGF2 neutralizing antibody to mice bearing experimental liver metastases phenocopied neutrophil depletion by reducing liver metastatic colony growth, vascular density, and branching (Hepatology 2017;65:1920-1935).

It has also been shown that FGFR1 inhibition by gene silencing in tumor cells or administration of a compound (SSR128129E at nanomolar concentrations) inhibiting FGF-induced signaling in tumor xenografts increases the efficiency of radiotherapy (pubmed/26896280).

Thus, an alternative possibility to explain the authors' results is that FGF2^{low} depletion directly causes initial TC death in the tumors in vivo, and that the resulting danger signals invite inflammatory/immune cells to the tumors. This possibility is consistent with the “T-cell independent alteration” that “resulted in less rapid tumor growth in FGF2^{low} mice than in WT mice”. Although the authors show that anti-FGF2 antibody treatment after irradiation did not alter clonogenic survival of SW1222 and HT29 cells, it would be essential to also analyze the effect of FGF2 silencing.

We appreciate the suggestion to use silencing of FGF2 in the cell lines, but would like to point out that because the different FGF2s are generated from different translational start starts, silencing of FGF2 would eliminate both the low molecular weight form and the high molecular weight FGF2s.

Thus we did not choose to use this strategy. We can point to other data that make this possibility unlikely. First if FGF2^{LMW} was behaving as an essential factor for tumour cells, the removal of T cells by the use of anti CD3 antibody in the *Fgf2*^{LMW^{-/-}} mice should not have made a difference as FGF2^{LMW} was still lacking (Fig 1I). In fact T cell depletion restored tumour growth. Second, the application of the antibody without radiation might have been expected to alter tumour growth. It did not.

To address this point in another way, have now added data to the manuscript showing that inhibition of all of the FGFRs with a pan FGFR inhibitor does not alter the clonogenic survival after radiation or the plating efficiencies of the three cancer cell lines used here. Overexpression of FGF2^{LMW} also did not alter clonogenic survival. In addition we did not find that blocking or addition of rFGF2 led to any change in the phosphorylation of AKT or of ERK that would be consistent with signalling by FGFRs in these cells (Supplemental Fig, 7).

In the manuscript, a limited analysis is reported on the role of TC produced FGF, and the expression of FGF receptors. A comparison of the amounts of FGF2 produced by TCs and macrophages should be shown.

Fig. 3B now compares FGF2 expression in BMDM, TAMs and the cancer cells MC38, SW1222 and HT29. As you can see TAMs have several hundred-fold higher expression than any of the others.

A more careful mapping of the autocrine pathway in the TCs is also needed.

We have looked for evidence of FGF2 triggered signaling through AKT and ERK and failed to find such evidence in tumour cells (new Fig. 3B). Unfortunately the current state of the art does not allow simple identification of phosphorylation of the FGFRs. We also tried and failed to obtain clean data with either IP of FGFRs or use of phosphospecific anti FGFR antibodies in Western blotting or immunohistochemistry. After speaking to others in the field, we found that our experience is typical.

It would be essential to compare the survival of the control, FGF2 and FGFR1 silenced as well as FGF2 antibody treated tumor cell cultures after irradiation.

As we noted above FGF2 silencing also affects the HMW forms which have extensive effects. Further the other FGFRs in addition to FGFR1 also have signalling roles so silencing one would not be sufficient. And other FGFs can also trigger FGFR1 and the others. In addition each of the cancer cell lines has a very different array of FGFRs. To address this point we added Supplemental Fig 7B which shows the varying levels of FGFRs in the tumour cell lines. As was noted above we also examined the tumour cell cultures after using a pan FGFR inhibitor without being able to detect any effects.

On pages 5 and 6 the authors show that T-cells are required for tumor regression in the FGF2^{lmw^{-/-}} mice. However, later, in Figure 5 F-H and Supplementary Figure 4, they show that T-cells are not required for the effects of the anti-FGF2 antibody on human SW1222 and HT29 tumor xenografts in immunodeficient mice. The authors should analyze the mouse tumor cell lines in using irradiation with and without the anti-FGF2 antibody in an immunocompromised background. Alternatively, they could use another inhibitor of the FGF2-FGFR pathway, such as SSR128129E, a FGFR1 blocking antibody, soluble receptor or FGFR1 gene silencing in the tumor cells.

Unfortunately we do not have sufficient remaining antibody to conduct any further in vivo experiments. There are 22 FGFs all of which can signal through the FGFRs. In addition to FGFR1, FGFR2 and 3 also have high affinity for FGF2. We believe it is beyond the scope of the study to investigate the effects of signalling through each FGFR. This would be in fact further complicated by the multiple splice variants of each receptor. The antibody to FGF2 is specific and other means of blocking FGFRs would not clearly duplicate its specific effects.

In addition, the authors should discuss why anti-FGF2 as a monotherapy does not decrease xenograft growth (Fig 5 F-H), although FGF2^{LMW} deletion decreases tumor growth in the isogenic mice (Fig. 1).

We have investigated this interesting question. We believe that this was due to different inocula used to generate the allografts. To address this point, we examined the effect of an increased inoculum of MC38 cells. We had used different inocula in these experiments. As noted in Fig. 1C at 1×10^4 cells, the tumours regressed. In contrast at 2×10^5 cells they did not (Fig 7A). After this inoculum the antibody had no effect on tumour growth unless it was given in combination with radiation. As noted now in the text it has been previously reported that inocula of fewer cells is more likely to elicit an immune response (ref 38). With the tumours generated from higher inocula, radiation had a greater effect in FGF2^{LMW} mice than the antibody. It is not surprising that an antibody which is given for a limited period and may not fully block all FGF2^{LMW} has a lesser effect than the genetic elimination.

As the authors indicate in the introduction, FGF2^{LMW} is a highly potent angiogenic molecule. However, here the authors don't provide any tumor vessel analysis. This should be analyzed to exclude the possible role of decreased vascular density to the decreased tumor growth. If anti-FGF2 antibody treatment decreases vascular density as shown previously in the case of liver metastasis, it could explain why anti-FGF2 antibody treatment decreases SW1222 and HT29 tumor growth in immunodeficient mice.

We appreciate this point and have added an analysis of vascular density from each of the conditions as now shown in Supplemental Fig. 8. We now add to the text these comments and conclusions from this experiment. "...the data was model specific. FGF2 blocking antibody reduced vascularity slightly in the SW1222 model, but not in the MC38 model (Supplementary Fig. 8). In neither case did it affect tumour growth (Fig. 7 A-C). Radiation reduced vascularity in both models. Vascular density was not significantly reduced by FGF2 blocking antibody in irradiated SW1222 or MC38 tumours (Supplementary Fig. 8) Further tumours in Fgf2^{LMW} mice had approximately the same vascular density as tumours in WT mice. Irradiation reduced the vascular density in Fgf2^{LMW} mice (Supplementary Fig. 6B&D). Thus whilst anti-angiogenic effects might play a role, this effect did not appear to be substantial or generalizable."

Specific comments:

Another question concerns the specificity of the effects reported here. The authors should perhaps mention that the other 21 members of FGF family are not expressed or of concern here. Similarly, they should describe which of the four receptors are present in the tumor cells they use.

We thank the reviewer for this point and have added a sentence to the discussion indicating the other FGFs. It is beyond the scope of this study to analyze the other 21 FGFs. We added the text "It should be noted that there are at least 22 human FGFs. Blocking only FGF2 is likely to have very different effects than blocking less specifically the FGF-FGFR signalling pathways." As noted

above in response to reviewer 4, Supplemental Fig 7B was added to show the varying levels of FGFRs in the tumour cell lines.

When reporting immune infiltration to the metastases/tumors, the authors should also report if this concerned normal tissues or circulating immune cells.

All of the data was from material purely in tumour associated areas. We have added this to the method sections by noting that all flow cytometry data was from tumours.

The authors should re-isolate the escape clones that resumed growth and test if they regrow in corresponding experiments in FGF2^{lmw}^{-/-} mice.

This would be a really great experiment, but is beyond our current scope and timeline.

The LLC growth curves should be added to Supplementary Figure 1.

We appreciate the suggestion and this has been done. See Supplemental Fig. 1B.

Did TAMs express FGFR3 or FGFR4? Was phosphorylation of FGFR1 correlated with FGF2^{lmw} expression or irradiation in TCs or TAMs?

We appreciate this suggestion and FGFR3 and 4 expression is now shown in Fig. 2Av. As noted above, FGFR1 or any FGFR phosphorylation is not really possible to detect in the absence of considerable overexpression. We tried using the putative phosphorylation specific antibodies and were unable to obtain acceptable data. After speaking to experts in this field, we understand that our experience is typical.

The anti-FGF2 antibody was used for the MC38 clonogenic assay in Supplementary Figure 5. However, only human tumor cells were used in the in vivo experiments. Why were the mouse tumor cells not used in the in vivo tumor experiments?

MC38, a murine tumour cell line was used in the in vivo experiments.

As the authors explain, FGF2 expression has been reported to be induced in irradiated human tumors. Thus, Figs 5A and B are of little value. In Fig. 5E, much of the macrophage and FGF2 staining appears to be eradicated by the irradiation. The authors should show the specificity of anti-FGF2 by staining macrophages of the corresponding tumors grown in FGF2^{lmw}^{-/-} mice.

We have altered the micrographs shown (now at higher magnification) in what was previously Fig 5 and now Fig. 6 to make identification of the individual components more apparent. This should make clear that FGF2 is not lessened in the irradiated tumours. Quantitative analysis of the images also confirms this. We appreciated the suggestion to stain the tumours grown in *Fgf2*^{LMW}^{-/-} mice. This is now shown in Fig. 6D and in fact neither the MC38 tumours nor the F4/80 positive cells stained for FGF2.

Minor comments

• Fig. 1 J and K are mixed on page 6?

This has been corrected.

- How was liver tumor burden measured?

The area of the macroscopic metastases was measured in macroscopic images with the area of the liver used for normalization in calculating the percentage.

- How was tumor volume calculated? The authors should study M1/M2 ratio in tumors beyond the (very small) volume of 500mm³.

This has been added to the methods. “Tumour sizes were measured every other day using calipers. Tumour volumes were calculated using the formula length × width²/2.” Unfortunately our animal license and the way it is administered under the local ethics committee does not allow us to go in any significant way beyond the volume of 500mm³.

- Fig. 1 B: Please show the invasion front in high magnification, with the identification of the tumor cells (by IHC).

We have now shown higher magnification images in which based on nuclear morphology the tumour cells are clearly evident. We have circled cells likely to be lymphocytes. (One of the authors is a pathologist and has confirmed these identifications.)

- In all experiments, the number of mice should be mentioned in each group. The individual tumor growth curves are difficult to interpret, for example in Fig. 5F.

This has been done and we have tried to more clearly label the individual growth curves (now Fig. 6).

Reviewers' Comments:

Reviewer #1:

Remarks to the Author:

Overall, the authors have incorporated many changes and the manuscript has been improved. One concern is regarding the experiments provided in response to the suggestion to demonstrate that FGF2 directly activates FGFR signaling in TAMs. The authors have provided data in Supplemental Figure 3 in which they treated macrophages with recombinant FGF2 and examined expression of inflammatory genes. They also treated cells with tumor cell conditioned medium followed by an FGF2 antibody. It is not clear whether appropriate controls were included. The source of recombinant FGF2 was not found. BMDMs are highly sensitive to even low levels of endotoxin, commonly found in most recombinant protein preparations, which can impact expression of inflammatory genes. In addition, it does not appear that an isotype control antibody was used in the antibody blocking experiments. Therefore, these data are difficult to interpret and do not provide convincing evidence for direct regulation of macrophage polarization by FGF2 via FGFR signaling.

There is also concern regarding the experiment provided in Supplemental Figure 7C. These experiments are ideally performed following serum starvation for short timepoints (5-30 minutes). Given the transient nature of FGF2-induced signaling it is unclear that analysis of signaling pathways 48 hours after treatment (in an undefined serum setting) provides meaningful/interpretable information. Furthermore, an appropriate positive control should also be included (such as a cell known known to respond to FGF2 stimulation). While it is agreed that FGFR phosphorylation can be difficult to detect, phosphorylation of FRS-2 is generally considered to be a relevant surrogate and is more specific than Akt and ERK.

Reviewer #3:

Remarks to the Author:

The Authors have provided adequate responses and revisions after the first round of reviews

Reviewer #5:

Remarks to the Author:

This is an interesting and clearly written paper. this reviewer has been asked to comment on whether the issues raised by reviewer 4 have been adequately addressed.

In the most part, the comments from reviewer 4 have been addressed. However for point 3 the authors couldnt get the antibodies to work, and the autocrine pathway requested was not possible to provide.

Point 5 was responded to by 'beyond the scope of this study', which is a little disappointing but probably does change the message of the paper

Reviewer 4 point 7 - for angiogenic responses an n=3 is not sufficient for statistical analysis and this n-number should be increased

The point about regrowing escape cells in FGF2^{low} was not done. Again this is a shame, because although it may not change the paper it would have been of interest.

Reply to reviewers for NCOMMS-18-01173A - "FGF2 Alters Macrophage Polarization, Tumour Immunity and Growth and can be Targeted during Radiotherapy"

Reviewers' comments:

Reviewer #1 (Remarks to the Author):

Overall, the authors have incorporated many changes and the manuscript has been improved. One concern is regarding the experiments provided in response to the suggestion to demonstrate that FGF2 directly activates FGFR signaling in TAMs. The authors have provided data in Supplemental Figure 3 in which they treated macrophages with recombinant FGF2 and examined expression of inflammatory genes. They also treated cells with tumor cell conditioned medium followed by an FGF2 antibody. It is not clear whether appropriate controls were included. The source of recombinant FGF2 was not found. BMDMs are highly sensitive to even low levels of endotoxin, commonly found in most recombinant protein preparations, which can impact expression of inflammatory genes.

The rFGF2 was purchased from Biolegend, USA. Cat.710304. We thank the reviewer for pointing out the absence of this information that we have now added to the Methods section. The technical data sheet for this product indicates that it contains less than 0.01 ng of endotoxin per μg of protein. In addition we treated BMDM with LPS- an endotoxin, and with rFGF2. LPS stimulated pAKT but not pERK. However rFGF2 did not stimulate either suggesting that it was not behaving in a fashion similar to endotoxin (Fig. 1). We did not expect these cultures to respond to FGF2 because the BMDM lack FGFRs unless further stimulated for example by exposure to tumor conditioned medium. pERK was detected in naïve FGF2^{-/-} BMDM indicating that detection was not a problem. Hence we saw transcriptional responses only after pre-treatment with tumour-conditioned medium as described in the manuscript. We did not add these experiments to the manuscript.

Figure 1 Treatment of BMDM with FGFR inhibitor BGJ398, rFGF2, LPS and IL4

WT BMDM were treated with the indicated agents for 24h in serum-free conditions followed by Western blotting as indicated. The left rectangle indicates the naïve BMDM or those treated with rFGF2 and the right shows the LPS treated BMDM after treatment with FGF2. LPS led to increased pAKT ser473) but did not affect pERK. rFGF2 had no effect.

In addition, it does not appear that an isotype control antibody was used in the antibody blocking experiments.

We agree that this essential control is missing. Because our lab was shut down before we could redo these experiments and because they are not essential to the argument that FGF2 alters the expression profile of BMDM, we have chosen to remove Panels C and D which included the antibody.

Therefore, these data are difficult to interpret and do not provide convincing evidence for direct regulation of macrophage polarization by FGF2 via FGFR signaling.

In these experiments, it was our intention to show that exposure of BMDM (pre-treated with conditioned medium) to exogenous FGF2 would alter their expression of characteristic macrophage cytokines. We believe that we have shown this; that BMDM only when pre-treated with CM to induce FGFRs respond to rFGF2 by upregulating some key cytokines.

There is also concern regarding the experiment provided in Supplemental Figure 7C. These experiments are ideally performed following serum starvation for short timepoints (5-30 minutes). Given the transient nature of FGF2-induced signaling it is unclear that analysis of signaling pathways 48 hours after treatment (in an undefined serum setting) provides meaningful/interpretable information.

We would point out that others have found the maximal response to FGF2 to occur between 24-48h after exposure of cancer cells or endothelial cells to FGF2 (1-4). It is a biphasic response similar to many other growth factors. We regret that we did not make it clear that the addition of rFGF2 was under serum-starvation conditions and have added this to the methods section.

1. The influence of fibroblast growth factor 2 on the senescence of human adipose-derived mesenchymal stem cells during long-term culture. Cheng Y, Lin KH, Young TH, Cheng NC. *Stem Cells Transl Med.* 2020 :518-530.
2. Two distinct signalling pathways are involved in FGF2-stimulated proliferation of choriocapillary endothelial cells: a comparative study with VEGF. Zubilewicz A, Hecquet C, Jeanny JC, Soubrane G, Courtois Y, Mascarelli F. *Oncogene.* 2001 :1403-13.
3. Fibroblast Growth Factor 2 lethally sensitizes cancer cells to stress-targeted therapeutic inhibitors. Dias MH, Fonseca CS, Zeidler JD, Albuquerque LL, da Silva MS, et al. *Mol Oncol.* 2019 :290-306.
4. The FGFR/MEK/ERK/brachyury pathway is critical for chordoma cell growth and survival. Hu Y, Mintz A, Shah SR, Quinones-Hinojosa A, Hsu W. *Carcinogenesis.* 2014 :1491-9.

Furthermore, an appropriate positive control should also be included (such as a cell known to respond to FGF2 stimulation). While it is agreed that FGFR phosphorylation can be difficult to detect, phosphorylation of FRS-2 is generally considered to be a relevant surrogate and is more specific than Akt and ERK.

Above we showed a positive response by BMDM in samples evaluated at the same time. We agree that FRS-2 would be more specific, but in the absence of any response by Akt or ERK by the cancer cells, specificity does not seem to be a concern.

Reviewer #3 (Remarks to the Author):

The Authors have provided adequate responses and revisions after the first round of reviews

Reviewer #5 (Remarks to the Author):

This is an interesting and clearly written paper. this reviewer has been asked to comment on whether the issues raised by reviewer 4 have been adequately addressed.

In the most part, the comments from reviewer 4 have been addressed. However for point 3 the authors couldnt get the antibodies to work, and the autocrine pathway requested was not possible to provide.

Point 5 was responded to by 'beyond the scope of this study', which is a little disappointing but probably does change the message of the paper

We agree that there are a number of interesting avenues for future work.

Reviewer 4 point 7 - for angiogenic responses an n=3 is not sufficient for statistical analysis and this n-number should be increased

In response to this suggestion, we have determined the vascular density in tumours from two different cell lines treated with or without XRT or antibody to FGF2 with n=5 for each group. We obtained the vascular density from immunohistochemistry of a section of the entire tumour stained with anti-CD31 antibodies. In the previous version of this manuscript we had included data about the volume of the perfused vasculature in two cell lines one with an n=3 and one with an n=5. In this submission we have replaced the previous data with the vascular density data based upon immunohistochemistry. With both sets of data the effect of FGF2 blocking antibody on the vasculature is slight and not generalizable for both cell lines.

The point about regrowing escape cells in FGF2Imw-/- was not done. Again this is a shame, because although it may not change the paper it would have been of interest.

Again, we agree that there are a number of interesting avenues for future work.

Reviewers' Comments:

Reviewer #1:

Remarks to the Author:

The authors have addressed reviewer concerns.